# BoHV-1-Vectored BVDV-2 Subunit Vaccine Induces BVDV Cross-Reactive Cellular Immune Responses and Protects against BVDV-2 Challenge

**DOI:** 10.3390/vaccines9010046

**Published:** 2021-01-13

**Authors:** Shafiqul I. Chowdhury, Katrin Pannhorst, Neha Sangewar, Selvaraj Pavulraj, Xue Wen, Rhett W. Stout, Waithaka Mwangi, Daniel B. Paulsen

**Affiliations:** 1Department of Pathobiological Sciences, School of Veterinary Medicine, Louisiana State University, Baton Rouge, LA 70803, USA; katrinpannhorst@aol.com (K.P.); pselvaraj1@lsu.edu (S.P.); xwen12@lsu.edu (X.W.); rstout1@lsu.edu (R.W.S.); dpauls1@lsu.edu (D.B.P.); 2Department of Diagnostic Medicine/Pathobiology, College of Veterinary Medicine, Kansas State University, Manhattan, KS 66506, USA; nsangewar@vet.k-state.edu (N.S.); wmwangi@vet.k-state.edu (W.M.)

**Keywords:** BoHV-1, quadruple mutant, vaccine vector, BVDV-2, glycoproteins E2 and Erns, BVDV subunit vaccine, granulocyte monocyte colony-stimulating factor (GM-CSF), vaccine efficacy

## Abstract

The bovine respiratory disease complex (BRDC) remains a major problem for both beef and dairy cattle industries worldwide. BRDC frequently involves an initial viral respiratory infection resulting in immunosuppression, which creates a favorable condition for fatal secondary bacterial infection. Current polyvalent modified live vaccines against bovine herpesvirus type 1(BoHV-1) and bovine viral diarrhea virus (BVDV) have limitations concerning their safety and efficacy. To address these shortcomings and safety issues, we have constructed a quadruple gene mutated BoHV-1 vaccine vector (BoHV-1 QMV), which expresses BVDV type 2, chimeric E2 and Flag-tagged Erns-fused with bovine granulocyte monocyte colony-stimulating factor (GM-CSF) designated here as QMV-BVD2*. Here we compared the safety, immunogenicity, and protective efficacy of QMV-BVD2* vaccination in calves against BVDV-2 with Zoetis Bovi-shield Gold 3 trivalent (BoHV-1, BVDV types 1 and 2) vaccine. The QMV-BVD2* prototype subunit vaccine induced the BoHV-1 and BVDV-2 neutralizing antibody responses along with BVDV-1 and -2 cross-reactive cellular immune responses. Moreover, after a virulent BVDV-2 challenge, the QMV-BVD2* prototype subunit vaccine conferred a more rapid recall BVDV-2-specific neutralizing antibody response and considerably better recall BVDV types 1 and 2-cross protective cellular immune responses than that of the Zoetis Bovi-shield Gold 3.

## 1. Introduction

The bovine respiratory disease complex (BRDC) remains a major economic problem for both beef and dairy cattle industries in North America and worldwide due to calf mortality, treatment expenses, and additional labor incurred. The United States Department of Agriculture National Animal Health Monitoring Service reported that BRD affects 12.4% of calves during the pre-weaning period, resulting in 22.5% calf mortality. Additionally, 5.9% of post-weaning animals are eventually diagnosed with BRD, causing 46.5% of the mortality during that period [1]. The detrimental economic impact of BRD on the American beef industry is even larger than in the dairy industry. It is the most expensive disease affecting feedlot cattle, and it is estimated to cause losses of approximately one billion dollars per year in the USA [2]. BRDC frequently involves an initial viral respiratory infection followed by a secondary bacterial infection, i.e., *Mannheimia haemolytica* (*M. haemolytica*). The initial viral respiratory infection creates a favorable condition for colonization of the lungs, usually by *M. haemolytica*, resulting in severe pneumonia and death of infected cattle, especially in the feedlots [3].

Among the respiratory viral agents implicated in BRDC, BoHV-1, and BVDV play significant roles because both viruses cause immunosuppression. BoHV-1 downregulates MHC-I [4,5], causes abortive infection and loss of CD4^+^ T lymphocytes [6], and interferes with the migration of lymphocytes and macrophages to the site of infection by counteracting chemokine activity [7]. BVDV causes leukopenia by infecting and killing lymphocytes and plasma cells [8]. Consequently, initial BoHV-1 and BVDV infections facilitate secondary bacterial infections that lead to death [8,9,10]. Furthermore, BoHV-1 causes lifelong latency in trigeminal ganglia (TG) with intermittent reactivation and nasal virus shedding [9,11], whereas BVDV causes persistently infected animals that shed large amounts of virus [12]. As a result, both viruses are maintained in the cattle population [13].

BoHV-1 encodes at least two immunosuppressive envelope proteins, U_L_49.5 and glycoprotein (g) G. The U_L_49.5, a non-glycosylated alphaherpesvirus gN homolog, transiently down-regulates MHC class I antigen presentation, which allows the virus to escape T cell recognition and clearance of the infected cells [4,5,14]. Similarly, BoHV-1 gG and its homologs in alphaherpesviruses bind to different chemokines secreted by the infected cells and interfere with activated migration of lymphocytes and neutrophils to the site of infection [7]. Consequently, gG disrupts chemokine gradients allowing survival of the infected cell. BVDV is also well-skilled in evading the host’s innate and adaptive immunity. Most viruses have only one possibility when they infect a host: either “hit and run” or “infect and persist”. The BVDV has mastered both strategies: (i) it counteracts innate immunity, primarily by inhibiting interferon production; (ii) it causes a transient leukopenia by infecting and killing the T-lymphocytes and macrophages; (iii) it doesn’t harm its persistently infected (PI), immunotolerant host for its survival and maintenance; (iv) the PI animals shed large amounts of virus, which infects naïve animals, usually subclinically, over a short time; and (v) it can mutate rapidly [15,16]. These properties of both viruses are retained in the current modified-live virus (MLV) vaccine strains [9,11,17].

Current vaccinations against these viral diseases utilize polyvalent vaccines containing BoHV-1, BVDV, and BRSV in two formats: MLV or killed virus (KV) [18] vaccines. Additionally, in the EU countries, live and killed BoHV-1 envelope glycoprotein E (gE) gene-deleted vaccines are mandated instead of the traditional MLV and KV vaccines [9,11]. Like the BoHV-1 wild-type (wt) virus, BoHV-1 MLV vaccines establish lifelong latency in the TG and cause nasal virus shedding following latent reactivation. Similarly, BVDV MLV vaccines can also persistently infect calves and have an added risk of mutating or reverting to virulence. Also, BoHV-1 and BVDV MLV vaccine strains retain the immunosuppressive traits of their respective parental wt strains [9,11,17]. Assessment of the effect of widespread BVDV vaccination over several decades is disappointing since this effort has failed to lower BVDV prevalence [15,19]. This failure is due to the unique biology of BVDV infection, which was not fully understood for a long time, and is still widely underestimated [20]. Together, these problems associated with the current vaccines have further complicated the BRDC epidemiology in the field and perhaps contributed to outbreaks of abortion and/or respiratory infections in the vaccinated animals [9,11,21,22,23,24].

While MLV vaccines’ safety is of concern, the efficacy of inactivated vaccines is not adequate because they do not induce a cellular immune response. In one instance, an effort to influence the cellular immune response of the BVDV inactivated vaccine resulted in bovine neonatal pancytopenia [25,26]. BoHV-1 glycoprotein E deleted [Δ] marker vaccine is safer than the MLV because it is not transmitted from the vaccinated to the non-vaccinated animals, rarely shed following latency reactivation, and the vaccinated animals are distinguishable from the infected animals. However, based on the protective efficacy measured by neutralization antibody titers following vaccination, the gEΔ marker vaccine was less efficacious than the traditional MLV, glycoprotein C (gC)-, and Thymidine kinase (TK)-deleted vaccines [27]. To improve the gEΔ marker vaccine’s vaccine efficacy, we engineered a BoHV-1 triple gene-mutated virus (BoHV-1 TMV). In the BoHV-1 TMV; (i) the coding sequences of the U_L_49.5 ectodomain residues 30–32 and the entire cytoplasmic tail residues 80–96 were deleted, and (ii) the coding sequences for the gE cytoplasmic tail residues 452–575 (372 bp), the gE-Us9 intergenic region and the entire Us9 ORF (541 bp) were deleted [28]. We compared the vaccine efficacy of the BoHV-1 TMV with that of a gEΔ virus against a virulent BoHV-1 wt challenge and determined that its protective efficacy was significantly better than the gEΔ vaccine [28] while retaining the safety and serological marker properties of the gEΔ virus [29]. Our results showed that both after vaccination and challenge, BoHV-1 TMV generated a considerably better cellular immune response in calves. After the BoHV-1 wt challenge and compared with that of the sham- and gEΔ-vaccinated calves, BoHV-1 TMV-vaccinated calves; (i) had more rapid and significant increases in neutralizing antibody titers and (ii) had a markedly reduced and shorter duration of nasal virus shedding [28].

In the present study, our goal was to use the BoHV-1 TMV as a delivery vector for a BVDV subunit vaccine. Initially, we generated a construct designated BoHV-1 TMV-BVDV.E21, encoding a BVDV-1 envelope glycoprotein E2. However, BoHV-1 TMV-BVDV-1.E2 vaccine virus was less effective than a commercial MLV (Bovi-Shield Gold^®^ IBR-BVD; Zoetis) against a virulent BVDV-1 challenge, and thus, it needed improvement. Therefore, we additionally deleted the chemokine binding, gG envelope protein. In the resulting quadruple gene-deleted BoHV-1 (BoHV-1 QMV), the gG-dependent blockade of chemokine signaling for immune evasion was eliminated. The modified BoHV-1 QMV vector was used to generate novel construct whereby the chimeric genes encoding BVDV-2 E2 and Flag-tagged Erns-bovine granulocyte-macrophage colony-stimulating factor (GM-CSF) fusion (Erns-GM-CSF) proteins were inserted in the gE cytoplasmic tail (CT)-Us9 and gG deletion loci, respectively. GM-CSF is known to enhance both humoral and cellular immune responses in viral vaccines [30]. The results presented here demonstrate that BoHV-1 QMV expressing the BVDV-2 E2 and Erns-GM-CSF (QMV-E2/Erns-GM-CSF), hereafter designated as QMV-BVD2*, is a safe and effective vaccine for the protection of calves against BVDV-2. The QMV-BVD2* prototype subunit vaccine induced the BoHV-1 and BVDV-2 neutralizing antibody responses along with BVDV-1 and -2 cross-reactive cellular immune responses. Moreover, after a virulent BVDV-2 challenge, the QMV-BVD2* prototype subunit vaccine conferred a more rapid recall BVDV-2-specific neutralizing antibody response and a considerably better recall BVDV types 1 and 2-cross protective cellular immune responses than that of a commercial MLV trivalent (BoHV-1, BVDV-1 and -2 strains) MLV (Zoetis Bovi-shield Gold 3).

## 2. Materials and Methods

### 2.1. Cells

The Madin Darby bovine kidney (MDBK) cell line was maintained in Dulbecco’s modified Eagles medium (DMEM # 10-017-CV, Corning^®^, Corning, NY, USA) supplemented with 10% heat-inactivated fetal bovine serum (FBS; EquaFETAL, Atlas Biologicals, Fort Collins, CO, USA) and 1× antibiotic/antimycotic solution (cat. # 30-004-CI; Corning^®^).

### 2.2. Viruses

BoHV-1 wt Cooper (Colorado-1) strain was obtained from the American Type Culture Collection (#VR-864, ATCC^®^, Manassas, VA, USA), and low passage viral stocks were maintained at −80 °C. BoHV-1 TMV was generated previously [28]. The cytopathic (cp) BVDV-1a Singer strain was received from LSU Louisiana Animal Disease Diagnostic Laboratory (LAADL). BVDV-1b cp strain TGAC was received from Dr. C. Chase from South Dakota State University [18]. BVDV-2a (cp) strain 125 was kindly provided by Dr. Clayton Keling, the University of Nebraska, at Lincoln, Nebraska. BVDV-1b non-cytopathic (ncp) strain CA04011866a (designated hereafter as CA), and ncp BVDV-2a strains 890 and 1373 were obtained from USDA/APHIS, Ames, IA, USA.

### 2.3. Antibodies

BVDV types 1 and 2, E2-specific monoclonal antibody (mAb; # 348) and BVDV-2 E2-specific mAb (# BA-2) were from vmrd^®^ (Pullman, WA, USA). Anti-Flag-specific mAb (# F1804) was from Sigma-Aldrich^®^ (St. Louis, MO, USA). Donkey anti-mouse highly cross-absorbed secondary antibody conjugated, Alexa Fluor 488 (# A-21202) was from Invitrogen (Carlsbad, CA, USA).

### 2.4. Virus Titrations

Virus titration, in the cases of BoHV-1 and cytopathic BVDV-2 strain 125 (125) was performed by plaque assay. Each viral stock solution was serially diluted ten-fold in DMEM supplemented with 5% FBS and 1× antibiotic/antimycotic solution. 200 µL of each virus-dilution was applied in duplicate onto the wells of 24-well cell culture plates over confluent MDBK cells. The plates were incubated for 2 h at 37 °C in a CO_2_ incubator before cells were overlaid with 1.6% carboxyl methylcellulose (CMC-high viscosity, Sigma-Aldrich^®^, # C5013) in DMEM. After 48 h (BoHV-1) and 72 h (BVDV-2), cells were fixed with 10% formalin solution for 1 h at room temperature (RT) and stained with 0.35% crystal violet. Plaques were counted under a surgical microscope. Virus titer was expressed as plaque-forming units (PFUs)/mL by using the following calculation: Reciprocal of the highest virus dilution × average number of plaques (5–20 plaques) counted in the two wells × 5. The viral plaque assay of BVDV-2 (ncp) strain 890 was performed similarly as above (for 125), but the cells were fixed at RT for 20 min (3% paraformaldehyde solution in PBS), and the viral plaques were visualized by immunofluorescence [31] assay using the BVDV-specific (both types 1 and 2) mAb # 348 (vmrd^®^).

### 2.5. Construction of BoHV-1 QMV Vector Virus

BoHV-1 TMV was constructed earlier, in which (i) UL49.5 residues 30–32 and CT residues 80–96 were deleted and (ii) the entire gE CT-Us9 coding regions were deleted (Figure 1A–C). To further improve the immunogenicity of BoHV–1 TMV, we deleted/null–mutated the gG gene in the BoHV–1 TMV genome and generated a BoHV–1 QMV. The BoHV–1 gG (Us4) is flanked by Us3 and Us6 (gD) genes, on the left and right, respectively (Figure 1D). The nucleotide numbers are shown in Figure 1D, which corresponds to the GenBank accession number J×898220.

Earlier, we could not isolate a viable gG ORF–deleted virus when the entire gG ORF coding region was deleted. We suspected that the putative gD gene promoter sequence (an essential viral gene) might be partially overlapping with the gG ORF sequences at the carboxy end. Alternatively, it could be that the deletion might have affected the shared Us3/Us4 Poly A site (Figure 1D), situated down–stream of the gG stop codon (nt 118640–118645). Therefore, our strategy was (i) to delete only the coding region of amino–terminal 67 amino acids, including the start codon of gG (Figure 1D); and (ii) to incorporate a synthetic Poly A, replacement, sequence in the deletion site to compensate for the authentic Us3 Poly–A, and the KpnI and HindIII restriction sites (Figure 1D) for the insertion of a BVDV–2 Erns–GM–CSF chimeric gene (see below). We designed a 2055 bp long sequence, containing 5’–3’ as follows: The 1000 bp US3 (partial) and US3–gG intergenic sequence (nt 116260–117259) with a NotI restriction site at the 5’ end, followed by 34 bp sequence containing 5’–3’, a replacement Us3 PolyA sequence as it appears in the BoHV–1 genomic sequence with its 3 bp flanking sequence on either side (nt 1186637 to 118648), a KpnI restriction site, a 10 bp spacer sequence and a HindIII restriction site. The HindIII restriction site is flanked on the right by the 1012 bp carboxy–terminal gG ORF sequence (nt 117460–118472) followed by a NsiI restriction site at the 3’ end (Figure 1D). According to this configuration of the designed 2055 bp long sequence, a 200 bp gG ORF sequence coding for the gG amino–terminal 67 amino acids, including the start codon (nt 117260–117459) were deleted, and the downstream gG amino acid residues 68–444 (nt 117460–118620) were not translated. Further, the gG deletion locus was flanked by 1000 bp on the left and 1160 bp BoHV–1 genomic sequences on the right sides, respectively, for homologous recombination into the BoHV–1 genome (Figure 1D). The 2055 bp NotI–Nsi fragment designed above was synthesized and cloned into the corresponding sites of the plasmid pBME–amp (Biomatik Corporation, Kitchener, ON, Canada). The resulting plasmid clone’s integrity, pgGΔ, was verified by sequencing (Biomatik Corporation). A BoHV–1 QMV was generated by cotransfection/ homologous recombination of pgGΔ with the full–length BoHV–1 TMV DNA using Lipofectamine (Invitrogen) as described earlier [32]. Several putative BoHV–1 QMV recombinants were analyzed by sequencing the genomic region spanning the Us3–gD genes. One of the recombinants was selected for the insertion of the BVDV–2 E2 and Erns–GM–CSF chimeric genes (Figure 2).

### 2.6. Incorporation of Chimeric BVDV 2–E2 Gene Cassette in the BoHV–1 QMV Genome to Generate BoHV–1 QMV–BVD2.E2 

#### 2.6.1. Construction of BVDV–2 E2 Insertion Plasmid

The plasmid pgE CTΔ–Us9Δ was generated previously [28]. Briefly, it contains a 2400 bp BoHV–1 genomic sequence inserted into the EcoRI–HindIII sites of plasmid pGEM–7Z (Figure 1B). The EcoRI–HindIII fragment consisted of gE ORF sequence, nt 121595–122989 bp coding for 451 gE amino–terminal, gE Ecto– and gE transmembrane amino acids flanked by EcoRI and KpnI at the 5’ and 3’ ends, respectively, were fused to 1004 nt of partial BoHV-1 infected cell protein (bICP) 22 gene sequence flanked by KpnI and HindIII at the 5’ and 3’ ends respectively (Figure 1B). In this configuration, a 1004 bp BoHV–1 genomic sequence (nt 122989–123993) comprised of gE CT amino acids 452–575 (approx. 372 bp), 106 bp gE–Us9 intergenic region, the entire Us9 ORF coding sequence (435 bp), and 88 bp of the Us9–bICP22 intergenic region were deleted (GenBank accession # JX898220). Further, a 21 bp sequence containing four stop codons and a Poly A signal, as well as a KpnI restriction site, was incorporated in the gE CT–Us9 deletion locus (Figure 1B).

To construct a BVDV–2 E2 insertion plasmid, first, a 2806 bp BVDV–2 E2 chimeric gene cassette (pBVD2–E2 gene cassette) was synthesized (GenScript, Piscataway, NJ, USA), which consisted of the following: A 1286 bp sequence for human elongation factor– 1α (EF–1α) promotor flanked by KpnI (5’) and ClaI (3’) restriction sites, followed by a 1183 bp chimeric sequence containing, the Kozak sequence (*CGCCGCCACC*), BoHV–1 gD signal sequence (nt 118819 to 118875, #JX898220; aa 1–19, GenBank accession # AFB76672.1), and BVDV–2 E2 ORF coding sequence, codon–optimized for *Bos Taurus* (GenBank accession #AAC72814.1), followed by a 337 bp NsiI–KpnI fragment containing the V5 epitope, 6xHis coding sequence, a stop codon (TGA) and bovine growth hormone (BGH) Poly A sequence (Figure 2B; Appendix A). The chimeric BVDV–2 E2 gene cassette was cloned into the KpnI site of pUC 57, and the integrity of the inserted sequence was verified (GenScript).

The 2806 bp chimeric BVDV–2 E2 gene cassette was then recloned into the KpnI site of the pgE CTΔ–Us9Δ plasmid clone described above (Figure 2A). The resulting plasmid clone, pBoHV–1 gEΔCT US9Δ–BVDV2 E2–INS (pBVD2–E2.INS) (Figure 2D), contains the 2806 bp BVDV–2 E2 chimeric gene flanked by 1400 bp (on the left) and 1000 bp (on the right) BoHV–1 genomic sequences for recombination and incorporation of the BVDV–2 chimeric E2 gene into the gE CT–Us9 deletion site of BoHV–1 genome (Figure 2A,B,D). The integrity of the flanking BoHV–1 and the inserted chimeric E2 sequences was verified (Genelab, Pathobiological Sciences, Louisiana State University, Baton Rouge, LA, USA).

#### 2.6.2. Construction of BoHV–1 QMV–E2.2 (QMV–BVD2.E2) Virus by Homologous Recombination

To generate a BoHV–1 QMV–E2 virus, linearized pBVD2–E2.INS insertion vector DNA was transfected [32] with the full–length BoHV–1 QMV genomic DNA. Several putative recombinant viruses were identified by PCR. One putative BoHV–1 QMV–E2 recombinant was plaque purified, and the integrity of the flanking BoHV–1 genomic and the chimeric E2 gene sequences were verified by sequencing (Genelab, LSU).

### 2.7. Incorporation of Chimerc BVDV–2 Erns–GM–CSF.Flag Gene Cassette in the QMV–BVD2.E2 Genome to Generate QMV–BVD2– E2.Erns–GM–CSF.Flag (Designated Here After as QMV–BVD2*) 

#### 2.7.1. Construction of BVDV2–E^rns^–GM–CSF.Flag Insertion Plasmid

To construct a BVDV–2 E^rns^–GMCSF.Flag insertion plasmid (pBVD2–Erns*–INS), first a 2037 bp BVDV–2 Erns–GMCSF–Flag chimeric gene cassette (Figure 2C) was synthesized (Biomatik) as follows: A 605 bp human cytomegalovirus (HCMV) promotor sequence (nt 1 to 605, GenBank # CVU55763) with a KpnI restriction site at the 5’ end, followed by a 1183 bp nucleotide sequence containing a kozak sequence (*CGCCGCCACC*), BoHV–1 gD signal sequence (nt 118819 to 118875, # JX898220; aa 1–19, # AFB76672.1), codon–optimized (*Bos Taurus*) nucleotide sequences for BVDV–2 1373 Erns (aa 271 to 497, # AAD38683) and bovine GM–CSF, lacking the signal sequence (Methionine and residues 18 to 143, #NP776452), the nucleotide sequence coding for a flag tag (GACTACAAAGACGATGA–CGACAAG), a stop codon (TAA), the simian virus 40 (SV40) termination/polyadenylation (Poly A) site (nt 1411 to 1640, # CVU55763), and the restriction site for HindIII (Figure 2C; Appendix A). The chimeric BVDV2 ErnsGM–CSF.Flag gene cassette was cloned into the KpnI–HindIII restriction sites of plasmid pgGΔ (Figure 2A), and the integrity of the inserted sequence was verified (Biomatik). In the resulting plasmid clone pBVD2 Erns–GM–CSF.Flag insertion plasmid (pBVD2–Erns*–INS), the 2037 bp BVDV–2 Erns–GM–CSF chimeric gene sequence (Figure 2E) was flanked by 1000 bp on the left and 1160 bp BoHV–1 genomic sequences on the right sides, respectively. The integrity of the flanking BoHV–1 and the inserted chimeric Erns–GM–CSF sequences was verified (Genelab, LSU).

#### 2.7.2. Construction of BoHV–1 QMV–E2/Erns*–GM–CSF Virus (QMV–BVD2*)*

To generate a recombinant QMV–BVD2* vaccine virus, linearized pBVD–2 Erns*INS was cotransfected with full–length QMV–BVDV–2.E2 recombinant (constructed above in Section 2.6.2) genomic DNA. Two putative QMV–BVD2* recombinant viruses were plaque purified (3x) and verified by PCR and sequence analyses (Genelab, LSU). To determine the stability of QMV–BVD2*, the recombinant virus was passaged several times in MDBK cells and tested again for the expression of BVDV–2 E2 and Erns–GMCSF.Flag (Flag–tagged Erns–GMCSF). Low passage QMV–BVD2* viral stocks were maintained at −80 °C.

### 2.8. Mock–And Virus–Infected Cell Lysates, SDS PAGE, and Immunoblotting 

For Western blot analysis of chimeric E2 and Erns–GM–CSF expression by QMV–BVD2*, MDBK cells were mock–infected or infected with QMV–BVD2*, BoHV–1 wt, and non–cytopathic (ncp) BVDV–2 890. Cells were harvested by centrifugation/pelleting (2095× *g* at 4° C) in a Beckman Coulter centrifuge (Avanti J–54) after 24–36 h post–infection (hpi) when the cytopathic effect was 95–100% (QMV–BVD2* and BoHV–1 wt) or 5 days post–infection (BVDV–2 strain 890). Each cell pellet was resuspended in a lysis buffer (50 mM Tris, pH 8.0, 0.5 M NaCl, 5 mM EDTA, 1% Triton X–100 and 0.1% SDS) at 40% weight/volume to adjust/ normalize the differences between the amounts of the samples of cells and proteins. After 1 hr incubation of the cell lysates on ice, the solubilized proteins in the supernatant were collected by centrifugation (21,130× *g* for 15 min, at 4 °C). The supernatants were aliquoted and stored at −80° C. For SDS–PAGE/ immuno–blotting analysis, 25 μL of each cell lysate was mixed 1:1 with 2× sample buffer (250 mM Tris, pH 6.8, 4% SDS, 10% glycerol, 0.2% Bromophenol Blue and 2% β–mercaptoethanol), boiled for 5 min and loaded on the 10% SDS–PAGE gel. To detect the chimeric BVDV E2 (Figure 2B) and Flag–tagged Erns–GMCSF (Figure 2C) expressed by the QMV–BVD2* recombinant virus (Figure 2F), immunoblotting with the BVDV–2 E2–specific Mab (vmrd^®^, # BA–2) and anti–Flag–specific Mab (Sigma–Aldrich # F1804) respectively, was performed as described earlier [33].

### 2.9. Comparison of QMV–BVD2* Growth Characteristics with That of BoHV–1 wt in MDBK Cells 

To compare the growth characteristics of QMV–BVD2* with that of BoHV–1 wt, we determined average plaque morphologies and one–step growth curves of QMV–BVD2* and BoHV–1 wt. Two wells of a six–well plate containing a confluent monolayer of MDBK cells were infected with 80–100 PFU of QMV–BVD2* or BoHV–1 wt viruses and overlaid with 1.6% CMC at 2 h post–infection (2 hpi). At 48 hpi, the cells were fixed (10% formaldehyde) and stained with crystal violet. The average plaque size of wt and mutant viruses was determined by measuring approx. 50 randomly selected plaques for each virus under a microscope with a graduated ocular objective, as described earlier [14]. The one–step virus growth property of the QMV–BVD2* was compared with wt, as described earlier [34]. Virus titers were determined by standard plaque assay as described above in 2.4.

### 2.10. Animals and Experimental Design

Animal infection, handling, sample collection, and euthanasia protocols were previously approved by the LSU Institutional Animal Care and Use Committee (Protocol # 14-078). Fifteen, four to five–month–old cross–bred steer, bull, or heifer calves were obtained from a BVDV free supplier. Before inclusion in the study, the calves were tested for BoHV–1 and BVDV serum neutralizing (SN) antibody titers (4–<4) and nasal BVDV shedding to ensure BoHV–1/BVDV free status. Five calves with >4 BVDV–2 maternal SN antibody titers were selected for the control group. Five calves of the remaining 10 were allocated randomly to each of the two treatment groups. Group 1 (QMV–BVD* group) and group 2 (Bovi–Shield Gold^®^ IBR–BVD; Zoetis; designated hereafter as the “Bovi” group) were housed in pens located in the School of Veterinary Medicine (closed) large animal isolation barn. Two pens, holding either two or three calves from each of the two vaccine groups, were well isolated (more than 100 feet apart). Foot baths were located at the main entry and in front of the entrance to each pen. Five calves selected for the control group (group 3) had slightly higher maternal serum neutralizing titers (16–32). They were housed individually in separate pens at an open–air barn with a concrete floor and restricted access. The barn housing the control calves was approx. 100 yards away from the other barn, and a foot bath was located at the main entrance.

### 2.11. Vaccination and Challenge

Vaccination, challenge, and sample collection scheme are shown in Figure 3. After one week of acclimatization, QMV–BVD2* group was vaccinated both intranasally (IN) with 2 × 10^7^ PFU/nostril and subcutaneously (SQ) with 1 × 10^7^ PFU. The calves in the “Bovi” group were vaccinated SQ according to manufactures recommendations, and calves in the control group were sham vaccinated intranasally with 1.0 mL of cell culture media. The calves in QMV–BVD2* and “Bovi” groups received the Micotil^®^ 300 (Tilmicosin; 20 mg/kg body weight) by SQ injection to prevent secondary bacterial infection. At 34 days post–vaccination (dpv), animals of all groups were challenged intranasally with a total of 2 × 10^6^ PFU (1 × 10^6^ PFU/mL/nostril) of ncp BVDV–2 890. The animals received Microtil^®^ 300 subcutaneously as above. The experiment was terminated at 54 dpv/20 days post–challenge (dpc). Upon euthanasia, a complete necropsy was performed to investigate any gross pathological lesions and collect tissue samples.

### 2.12. Clinical Assessment of Calves

Calves were clinically assessed for the rectal temperature, feed, and water intake, on the day of vaccination (0 dpv) and 2, 4, 6, 9, 14, 21, and 28 dpv and on 34 dpv/0 dpc (Figure 3). Following the challenge with ncp BVDV–2 890, clinical signs were recorded daily until 14 dpc (Figure 3). The clinical assessment included rectal temperature, nasal and ocular discharge, dyspnea, coughing, lethargy, anorexia, mucosal and oral lesions, and diarrhea. Clinical scores for each animal were calculated based on the criteria listed in Appendix A.

### 2.13. Sample Collection and Processing

The schedule of EDTA–blood, serum, and nasal swab collection is shown in Figure 3. Nasopharyngeal swabs were collected in 1 ml of cell culture media, supplemented with 2× antibiotic/antimycotic solution, on 0, 2, 4, 6, 7, 9, 14, and 21 dpv, and 0, 2, 4, 6, 8, and 11 dpc (Figure 3). The samples were processed and stored at −80 °C. Virus titrations by plaque assay were performed as described above in Section 2.4.

### 2.14. Isolation and Freezing of PBMC

PBMCs were isolated using Ficoll–Paque (Ficoll–PaqueTM PLUS, GE healthcare, Chicago, IL, USA) density–gradient centrifugation as previously described [35]. For freezing, isolated PBMCs were resuspended in 10% FBS–RPMI–1640 medium containing 10% dimethyl sulfoxide (DMSO; Sigma–Aldrich) at a concentration of 5 × 10^6^ cells/ml. Aliquots of PBMCs were subjected to slow freezing at −80 °C (overnight) before transferring to a liquid nitrogen tank for long–term storage.

### 2.15. Leukocyte Counting in Whole Blood–EDTA Samples

For counting the leukocytes in whole blood–EDTA samples, an automatic hematological analyzer (Advia 120; Siemens Healthcare Diagnostics, Tarrytown, NY, USA) was used. On the day of challenge (0 dpc) and on 4, 6, 8, 11, and 14 dpc, total leukocyte counts were determined and recorded. In addition, the percent decline in leukocyte numbers in each calf was calculated as follows and described earlier [36]:(1)$$\begin{array}{c} \%\;\mathrm{decline}\;\mathrm{of}\;\mathrm{leukocyte}\;\mathrm{count}\;\mathrm{of}\;\mathrm{an}\;\mathrm{animal}\;“X”\;=\;100\\ \;-\frac{\mathrm{Lowest}\;\mathrm{leukocyte}\;\mathrm{count}\;\mathrm{of}\;\mathrm{an}\;\mathrm{animal}\;“X”}{\mathrm{Leukocyte}\;\mathrm{count}\;\mathrm{of}\;\mathrm{the}\;\mathrm{animal}\;\mathrm{on}\;0\;\mathrm{dpc}}\times 100 \end{array}$$

A decline in the leukocyte count of more than 25% was considered leukopenia [37].

### 2.16. Virus–Neutralization Assay

Sera were heat–inactivated at 56 °C for 30 min. 250 µL of BoHV–1 wt Cooper or BVDV–2 125 virus suspension containing approx. 100 PFUs/100 µL were preincubated with 250 µL of serial four–fold serum dilutions (for BVDV–2) or serial two–fold serum dilutions (for BoHV–1) at 37 °C for 2 h. Similarly, 250 µL of plain cell culture media was incubated with 250 µL of the respective virus suspensions in 6–8 tubes (virus control) and incubated at 37 °C for 2 h. 200 Two hundred microliters of the serum–virus mixture from each serum dilution were added to two wells (duplicate) of 24–well cell culture plates containing confluent MDBK cells. For the virus control, 200 µL of virus–media mixtures were added to 6–8 wells of 24–well plates. After 2 h incubation at 37 °C, 0.8 mL of 1.6% CMC in DMEM was added to each well. The plates were incubated in a CO_2_ incubator at 37 °C for two days for the BoHV–1 and 4 days for the BVDV–2 plaque assays. After fixing the cells with 10% formalin (1–2 h) and washing with tap water, the cells were stained with 0.35% crystal violet solution (20 min). The viral plaques in the serum–virus mixture wells and their respective virus control wells were counted under a microscope. The reciprocal of the highest dilution of each serum that inhibited/neutralized 50% of the average number of the respective control virus plaques, but not less than 40–45 plaques, was reported as the virus–neutralization titer.

### 2.17. Reverse Transcription Quantitative Real–Time PCR (RT–qPCR) for BVDV Viremia

To detect BVDV, we performed RT–qPCR on post–challenge PBMCs samples. Briefly, total RNAs were extracted from the PBMCs of calves at 0, 4, 6, 8, and 11 dpc, using RNA easy extraction kit (Qiagen^®^, Hilden, North Rhine-Westphalia, Germany) according to the manufacturer’s recommendations. cDNA was generated from 250 ng of total RNA followed by RT–qPCR using the VetMax–Gold BVDV detection kit (Thermo Fisher Scientific, Waltham, MA, USA, # 4413938). RNA isolation was performed two times, and the RT–qPCR analysis was repeated three times in duplicate for each sample. BVDV genome load was calculated following the manufacturer’s instruction. According to the manufacturer’s instruction, 1 µL of the positive control (25 × BVDV RNA) contains 10,000 copies of BVDV. To generate a standard curve, 8 µL of the positive control were serial diluted 10–fold. Standard samples corresponding to 4, 40, 400, and 4000 copies were included in each PCR analysis. BVDV copy numbers in each sample were calculated according to the standard curve’s CT–values, divided by 250 to BVDV genome in one ng total RNA. All samples that had a copy number lower than the highest copy number detected in samples from 0 dpc (Threshold of 2.32 copies/ng total RNA) were evaluated as negative.

### 2.18. BVDV–Specific Cellular IFN–γ and Proliferation Responses

At day 0 and 14 post–vaccination and day 6 post–challenge, IFN–γ responses in PBMCs were evaluated by enzyme–linked immunospot (ELISPOT) assay. The assay was performed using Bovine IFN–γ ELISpot^BASIC^ (ALP) kit (Mabtech, Stockholm, Sweden, # 3119–2A) as per the manufacturer’s instruction and as described previously [31,38]. Briefly, 0.25 × 10^6^ whole–blood–derived PBMCs were seeded in triplicate wells of MultiScreen–IP plates (MilliporeSigma™, # MAIPS4510) with whole heat–killed BVDV virus [CA0401186a (CA), TGAC, A125 or 1373] in a final volume of 100 µL complete RPMI 1640 medium. The positive control was 2.5 µg/ml concanavalin A (ConA), whereas medium alone was used as a negative control. The spots were quantified with an ELISPOT reader, Cellular Technology Limited (CTL, Shaker Heights, OH, USA) ImmunoSpot^®^ S6 Analyzer. The results were presented as the mean number of BVDV–specific IFN–γ^+^ spot–forming cells (SFC) per 10^6^ PBMCs after deducted background medium counts.

BVDV–specific PBMC proliferation responses on 14 dpv and 4 dpc were determined using ^3^H–Thymidine incorporation assay as previously described [31,38]. Briefly, 0.5 × 10^6^ whole–blood–derived PBMCs were cultured for 72 h at 37 °C in triplicate wells of round–bottom 96–well plates in a total volume of 100 μL of complete RPMI 1640 medium containing 10 µg/mL of whole heat–killed BVDV virus. The positive control was 1.25 μg/mLConA, whereas medium alone was used as a negative control. Cells were labeled with 0.25 μCi of ^3^H–thymidine for 12 h and then harvested using a semi–automatic cell harvester (Perkin Elmer, Waltham, MA, USA), and the incorporated ^3^H–thymidine was counted with a Micro–Beta liquid scintillation counter (Perkin Elmer). The incorporation of ^3^H–thymidine by the proliferating PBMCs was presented as mean counts per minute (CPM) of triplicate wells after deducting the background medium counts.

### 2.19. Euthanasia, Necropsy and Pathology

Calves were euthanatized with xylazine and pentabarbitol 20 dpc (Figure 3). Complete necropsies were performed. Samples were collected from the lungs, systematically from cranial and caudal lobes with additional sections from gross lesions. Lungs were photographed, and lesions were recorded. Samples were also collected from the kidney, liver, spleen, mesenteric lymph node, tonsil, bronchial lymph node, cervical lymph node, and bone marrow. Samples were fixed in 10% buffered–neutral formalin. Hematoxylin–eosin (HE) stained slides were obtained from paraffin–embedded tissues by routine methods. All tissues were evaluated and scored by a single veterinary pathologist, who was blinded to treatment.

Tissues, except lungs, were scored on a scale of 0–4 (0 = normal, 1 = minimal, 2 = mild, 3 = moderate, and 4 = severe) on multiple parameters. Parameters included acute inflammation, chronic inflammation, and necrosis for all tissues with additional tissue–specific parameters, such as glomerular changes for kidneys, lymphoid depletion, and hyperplasia for all lymphoid organs, sinus histiocytosis for lymph nodes, and myeloid/erythroid hyperplasia in the bone marrow. All lung sections were individually scored on a 0–3 scale (0 = normal, 1 = mild, 2 = moderate, and 3 = severe). Bonchi/bronchioles, parenchyma, and septa/pleura were evaluated in each section.

### 2.20. Statistical Analysis

#### 2.20.1. Nasal Virus Shedding and Viremia

All data were expressed as means ± standard deviation. Statistical analyses were performed using GraphPad PRISM^®^ software version 5.04 (GraphPad Software, San Diego, CA, USA). The two–way analysis of variance (ANOVA) followed by Bonferroni posts–tests to compare replicate means by row were performed. A value of *p* ˂ 0.05 was considered statistically significant.

#### 2.20.2. Calculation of Outliers

Outliers in data points that differ significantly from other observations were estimated by Grubb’s test (generalized extreme studentized deviate method) with alpha level of 0.05 using GraphPad PRISM^®^ software.

#### 2.20.3. Cellular Immune Response

The Nonparametric Kruskal–Wallis test with Dunn’s multiple comparisons test was used to analyze the significant differences between groups. Post–vaccination and post–challenge, the significance of the differences in BVDV–specific immune responses (cellular IFN–γ and proliferation responses) were compared among all groups. Statistical analysis was performed using GraphPad Prism 7 (Version 7.04, GraphPad Software). A significance level of <0.05 was used for all analyses.

#### 2.20.4. Histopathology of Lung Sections

A nonparametric rank test of factorial ANOVA with repeated measures was performed to detect the differences in the three vaccinated groups’ efficacy levels, adjusted by Aligned Rank Transformation [22]. The rationale of this approach is to allow us to perform nonparametric factorial analyses when handling repeated measures [39]. This approach is more robust to test sophisticated data structure than other traditional nonparametric tests [40]. The adjustment method aligned rank transform (ART) relies on alignment and ranking step before using F–tests. Therefore, ART is similar to the parametric ANOVA, except that the response variable may be continuous or ordinal and is not required to be normally distributed. Post–hoc pairwise comparisons were conducted, and the alpha levels were adjusted by Tukey method. All statistical analyses were performed by R.

## 3. Results

### 3.1. Characterization QMV–BVD–2* Recombinant Virus

#### 3.1.1. QMV–BVD2* Virus Expresses the Chimeric BVDV–2 E2 and Erns–GM–CSF.Flag Proteins

Sequence analysis of the Erns–GM–CSF and E2 chimeric genes and their flanking, QMV–BVD2* sequence (approx. 1000 bp on each side) validated the chimeric gene sequences’ integrity and their insertion at the gG and gECT–Us9 deletion loci, respectively. Further, the expression of chimeric BVDV–2 E2 antigen and Flag–tagged Erns–GM–CSF in QMV–BVD2*–infected cell lysates was verified by SDS–PAGE/Western immunoblotting. For this, QMV–BVD2*, BoHV–1 wt–and BVDV–2 890–infected cell lysates were tested for chimeric BVDV–2 E2 and Flag–tagged Erns–GM–CSF expression. A BVDV E2–specific mAb recognized an approx. 53–55 kD protein band both in QMV–BVD2* and BVDV–2 strain 890–infected cell lysates (Figure 4A), and the anti–Flag mAb recognized a 56 kD Flag–tagged Erns–GM–CSF band in the QMV–BVD2*–infected cell lysate (Figure 4A). As expected, the E2 mAb did not bind to any band in the BoHV–1 wt–infected cell lysate, and the anti–Flag mAb did not recognize any protein band in the BoHV–1 wt and 890–infected cell lysate (Figure 4A). As noted earlier, the E2 chimeric protein is fused to V5 epitope and 6x His tags at the carboxy end. The combined molecular mass of V5 and 6x His is approx. 2.2 kD, which was not large enough to make a noticeable shift of the chimeric E2 band of QMV–BVD2* compared to the authentic E2 band of the BVDV–2 strain 890 (Figure 4A). The estimated molecular mass (https://web.expasy.org/cgi–bin/compute_pi/pi_tool) of the chimeric Flag–tagged Erns–GM–CSF protein is 43.8 kD; Erns (25.7 kD), GM–CSF (17.13kD, and Flag (1.02 kD)). However, as noted above, the anti–Flag mAb detected an approx. 56 kD band on the Western blot containing the QMV–BVD2*–infected cell lysate (Figure 4A). The increase in the chimeric Erns–GM–CSF–Flag protein’s size is most likely due to glycosylation because the amino acid sequence analysis (NetNGlyc 1.0 server, Technical University of Denmark; http://www.cbs.dtu.dk/cgi–bin/webface2.fcgi?jobid) predicted at least six major, potential N glycosylation sites. Even though the same amounts of proteins were loaded in the two duplicate gels (2.8 in M & M), the amount of Flag–tagged chimeric Erns–GM–CSF protein detected by the anti–Flag antibody, on the Western immunoblot (Figure 4A) was considerably less compared with that of chimeric E2 recognized by the Anti–BVDV–2 E2–specific mAb. This is because BVDV Erns protein is secreted as well as membrane–anchored but the E2 is only membrane–anchored [41].

#### 3.1.2. QMV–BVD2* Virus Produces Smaller Plaques but Replicates with Similar Kinetics and Yield Compared with the BoHV–1 wt

The QMV–BVD2* virus produced smaller plaques than BoHV–1 wt (Figure 4B), and the plaques were very similar to those of the BoHV–1 TMV virus described earlier [28]. The one–step growth curve results show that the growth kinetics and virus yield of QMV–BVD2* is similar to that of BoHV–1 wt (Figure 4C).

### 3.2. Pathogenicity and Nasal Virus Shedding Following IN/SQ Vaccination with Live QMV–BVD2* Subunit Vaccine

Following immunization with QMV–BVD2* (intranasal/subcutaneous; IN/SQ) and “Bovi” (SQ), the calves remained clinically normal regardless of the vaccine used. As expected, 2 dpv, all the QMV–BVD2* vaccinated animals (5/5) shed the vaccine virus with an (average titer 2.26 × 10^2^ PFU/mL). On the 4 dpv, four animals (4/5) shed the virus (average titer 2.7 × 10^3^ PFU/ml) (data not shown). On 6, 7, and 9 dpv QMV–BVD2* vaccine virus could not be isolated from any of the QMV–BVD2* vaccinated animals. None of the “Bovi” vaccinees and the negative control calves shed any vaccine virus in the nose. One calf (# 648) in the QMV–BVD2* group developed diarrhea and fever on 28 dpv due to an unknown cause. The calf was treated with antibiotics and physiological saline infusion. The calf was later euthanized prior to the challenge.

### 3.3. Post–Vaccination BoHV–1 Serum Neutralizing Antibody Titers in the QMV–BVD2* Group Was Slightly Lower than That of the "Bovi Group, but BVDV–2 Neutralizing Antibody Titers in the “Bovi” Group Was Considerably Better than in QMV–BVD2*

On the day of vaccination (0 dpv), mean BoHV–1– and BVDV–2–specific maternal antibody titers in both QMV–BVD2* and “Bovi” vaccine groups were approx. 4 (Figure 5A,B). However, the control calves had a relatively high titer (approx. 31) against BVDV–2 (Figure 5B; Appendix A) but not against BoHV–1 (Figure 5A). Following vaccination, the average BoHV–1–specific neutralizing titers in the QMV–BVD2* treatment group rose to 9 (two–fold) by 34 dpv (0 dpc or on the day of the challenge) (Figure 5A). In the case of “Bovi” treatment group, the corresponding increase in the neutralizing titers was slightly more than three–fold (from 4 to 14) (Figure 5A).

As depicted in Figure 5B and Appendix A, the mean BVDV–2–specific maternal neutralizing antibody titers in the control animals declined to 10 by 34 dpv (0 dpc). In contrast, by 34 dpv, the BVDV2–specific neutralizing titers increased, four–fold, from 4 to 17 (seroconverted) in QMV–BVD2* group and the corresponding titers in the “Bovi” group rose 210–fold (from 4–878) (Figure 5B; Appendix A).

### 3.4. QMV–BVD2* Vaccinated Calves Induced Higher BVDV Cross–Reactive (Types 1 and 2) IFN–γ Responses than That of the “Bovi” Group

We determined IFN–γ responses in the PBMCs collected on 0 and 14 dpv against heat–killed BVDV–1 strains CA (ncp) and TGAC (cp), and BVDV–2 strains 1373 (ncp) and A125 (cp), and ncp1373) strains by ELISPOT assays (Figure 6). All the animals were negative on 0 dpv; however, on 14 dpv, BVDV–1– and –2–specific IFN–γ responses were detected in the calves vaccinated with either QMV–BVD2* or “Bovi”, but not in the negative controls (Figure 6A,B).

Notably, on 14 dpv, QMV–BVD2*–vaccinated calves had the highest mean IFN–γ responses against both the BVDV–1 CA (88) and TGAC (144*) strains (Figure 6A). Notably, the mean TGAC–specific IFN–γ response detected in the QMV–BVD2*–vaccinated calves (144*) were significantly higher (* *p* < 0.05) than the mean response seen in the negative control calves (Figure 6A). Similarly, post–vaccination BVDV–2 (125 and 1373)–specific IFN–γ responses in QMV–BVD2*–vaccinated calves were the highest among the three treatment groups (Figure 6B). In addition, mean responses detected in “Bovi” and QMV–BVD2*–vaccinated calves for the two BVDV–2 strains were significantly higher (* *p* < 0.05) than the mean responses in the negative control calves (Figure 6B).

### 3.5. QMV–BVD2* Vaccinated Calves Induced Higher Cross–Reactive (BVDV–1 and –2) Recall Cell (PBMC) Proliferation Responses than That of the “Bovi” Group

Recall cell proliferation responses against BVDV–1 and –2 were evaluated on day 14 post–vaccination (Figure 7A,B). Among the three treatment groups, the highest mean BVDV–1–specific cell proliferation response was detected in QMV–BVD2* vaccinees (Figure 7A). However, no significant difference was detected between the mean responses of the QMV–BVD2* vaccinees against two BVDV–1 strains compared with the mean response in the control group (Figure 7A). Nevertheless, the mean proliferation response of the QMV–BVD2* vaccinees was two–fold higher than the “Bovi” vaccinees against the TGAC strain (Figure 7A).

The QMV–BVD2* experimental vaccine also generated the highest mean BVDV–2 specific cell proliferation responses on 14 dpv against A125 and 1373 strains among the three treatment groups (Figure 7B). Although the QMV–BVD2* responses compared with that of the “Bovi” group were 2–fold higher, the proliferation responses were not significant relative to the control group (Figure 7B).

### 3.6. QMV–BVD2* Vaccinated Calves Induced Higher Memory Serum Neutralization Antibody Response after Challenge with BVDV–2 than That of the “Bovi” Group

Following challenge with ncp BVDV–2 890, average neutralizing antibody titer in control calves decreased from 10 on 0 dpc to 6 on 6 dpc, which was most likely the decline in maternal antibody titer (Appendix A). In the commercial “Bovi” vaccine group, the average neutralizing titer against BVDV–2 did not rise after the challenge at 6dpc (increased from 878 at 0 dpc to 885 on 6 dpc) or did not seroconvert (Figure 5B, Appendix A). In contrast, in the QMV–BVD2* vaccinated group, the BVDV–2–specific average neutralizing titer increased from 17 at 0 dpc to 64 at 6 dpc, a 3.8–fold (four–fold) increase or seroconverted (Figure 5B, Appendix A). Remarkably, at 14 dpc, the average neutralizing titer in the QMV–BVD2* group was 94,208, a 5500–fold increase compared to 0 dpc (Figure 5B; Appendix A). In contrast, the average neutralizing titer of the “Bovi” and sham (control) vaccinated groups, on 14 dpc, were 16,589 and 1638, which were six– and 58–folds less than that of the QMV–BVD2*, respectively (Figure 5B; Appendix A). Therefore, upon BVDV–2 challenge, the relative recall serum neutralizing antibody responses of the QMV–BVD2* vaccinated calves at 14 dpc was more robust than that of the commercial “Bovi” vaccinated calves.

### 3.7. QMV–BVD2* Vaccinated Animals Had the Highest BVDV Cross–Reactive Post–Challenge Cellular IFN–γ and Proliferation Responses

At 6 days post–challenge, PBMCs from the QMV–BVD2* vaccinated animals had the highest mean BVDV–1– and BVDV–2–specific recall IFN–γ responses amongst the three treatment groups (Figure 6). However, the differences among the groups were not significant. (Figure 6A,B). Overall, the mean IFN–γ responses recalled post–challenge in the QMV–BVD2*–vaccinated animals were at least two–fold higher than that of the “Bovi” vaccinees against BVDV–1 and –2 strains (Figure 6A,B).

Notably, upon BVDV–2 challenge, QMV–BVD2* experimental vaccine generated the highest mean cross–reactive (BVDV–1– and –2) recall cell proliferation responses among the three treatment groups (Figure 7A,B). Specifically, in comparison to the negative controls, the QMV–BVD2* vaccinees had significantly higher post–challenge recall cell proliferation against both BVDV–1 strains: CA (* *p* < 0.05) and TGAC (** *p* < 0.01) (Figure 7A) and against BVDV–2 strain 125 (** *p* < 0.01) (Figure 7B). Although the proliferation responses recalled in the QMV–BVD2*–vaccinees against BVDV–2 strain 1373 was the highest, the mean proliferation level compared with that of the control calves was not significant (Figure 7B). Nevertheless, the mean cellular proliferation response in the QMV–BVD2* treatment group was at least two–fold higher than that of the “Bovi” treatment group against the 1373 strain (Figure 7B).

### 3.8. QMV–BVD2* Vaccinated Calves Had a Mild and Brief Period of Leukopenia after the BVDV–2 Challenge

On the day of challenge (0 dpc) and on 4, 6, 8, 11, and 14 dpc, total leukocyte counts were determined and recorded (Figure 8A). Based on the criteria used to calculate the percent decline in leukocyte numbers in material and methods, all control animals had leukopenia from 4 dpc until 14 dpc. Notably, the highest percentage decline in the control animals ranged between 29–56%, with a 47% mean percentage decline in the group (Figure 8B). Based on the percent decline in leukocyte count, three calves from the QMV–BVD2* treatment group had mild leukopenia (approx. 30% decline, which is 5% more than the threshold) while calf # 630 had a drop of 13%, which lasted for three days, 4 dpc–6 dpc (Figure 8B). None of the “Bovi” vaccinated calves had leukopenia (Figure 8B). In the QMV–BVD2* treatment group, there was a brief drop in leukocyte counts on 6 dpc from 10.2 × 10^3^ to 8 × 10^3^/µL (Figure 8A), which was still within the average values. However, the mean leukocyte counts rose back to 9.5 × 10^3^/µL by 6 dpc in three of four calves (Figure 8A). Notably, calf # 630 had a low leukocyte count of 6.3 × 10^3^/µL on 0 dpc (Appendix A), which is at the lower end of a normal range (6–12 × 10^3^) and 2 × 10^3^ fewer than the mean, standard leukocyte count of 8 × 10^3^. This calf had leukopenia both on 4 dpc and 6 dpc, but the number rose back to 10 × 10^3^/µL on 8 dpc (Appendix A).

### 3.9. Except for One Outlier in the QMV–BVD2* Group, Both QMV–BVD2* and “Bovi” Vaccinated Groups Did Not Shed the Challenge Virus in the Nasal Secretion

On 4 dpc, three control animals shed low (5–40 PFU’s/mL) to moderate amounts (1.8 × 10^2^ PFU’s/mL) in the nose (Figure 9A; Appendix A). On 6 dpc and 8 dpc, all five control animals shed virus, and one animal (# 638), shed relatively high amounts of the virus on 6 dpc and 8 dpc, 5.5 × 10^2^ PFU’s/ml and 1.25 × 10^2^ PFU’s, respectively (Appendix A). On 11 dpc, only one control animal (# 637) shed a low amount of virus (5 PFU’s/mL). In the case of QMV–BVD2*, only one (# 630) of four animals shed a few amounts of challenge virus on 4 dpc (5 PFU’s/mL) and 6 dpc (35 PFU’s/mL) (Figure 9A; Appendix A). Notably, calf # 630 also had the lowest leukocyte count among all the groups on the day of the challenge (6.9 × 10^3^ ), which is at the lower end of a normal range (6–12 × 10^3^) and 2 × 10^3^ fewer than the mean (Appendix A).

None of the “Bovi” vaccinees shed the challenge virus at a detectable level by plaque assay. With the exception of one calf (# 630), the QMV–BVD2* vaccinees also did not shed the challenge virus. Nevertheless, as a whole, compared with the sham–vaccinated control calves, nasal virus shedding in the QMV–BVD2*–vaccinated calves were reduced significantly (Figure 9A; Appendix A).

### 3.10. Except for One Outlier in the QMV–BVD2* Group, Both QMV–BVD2* and “Bovi” Vaccinated Groups Did Not Have Viremia upon Challenge

BVDV viremia was assessed by detecting BVDV genomic copies in PBMCs by RT–qPCR. As depicted in Figure 9B, all control animals had viremia for 8 days, starting on 4 dpc until 11 dpc (the last day of assessment). The average copy number of the BVDV genome/ng total RNA determined for the unvaccinated control animals was 1.36 (0 dpc), 13.3 on 4 dpc, 272.26 on 6 dpc, 677.6 on 8pc, and 47.8 on 11 dpc (the last test performed)(Figure 9B; Appendix A). In the case of QMV–BVD2* vaccinated calves, the corresponding mean genome copy numbers were1.02 (0 dpc), 5.88 (4 dpc), 57.97 (6 dpc), 6.52 (8 dpc), and 1.73 (11 dpc) (Figure 9B; Appendix A). For the “Bovi” treatment group, the corresponding genome copy numbers were 1.16 (0 dpc), 1.71 (4 dpc), 2.29 (6 dpc), 2.28 (8 dpc) and 1.78 (11 dpc) (Figure 9B; Appendix A). However, when the genomic copy number in the individual calves belonging to the “Bovi” and QMV–BVD2* was analyzed, it was revealed that with the exception the calf # 630, which had 13.93, 57.97, and 20.04 BVDV genomic copies on 4, 6, and 8 dpc, respectively, the rest of the three calves in the group had much lower genomic copy numbers (Appendix A). Notably, during 4dpc–11 dpc, the genomic copy numbers in the PBMC’s of the other three calves in the QMV–BVD2* group were comparable to those in the “Bovi “treatment group (Appendix A). As noted above, animal # 630 (QMV–BVD2* group) also had a low leukocyte count (6.9 × 10^3^/µL) on the day of the challenge (Appendix A). Therefore, as a whole, compared with the sham–vaccinated control calves, the reduction in viremia in both the QMV–BVD2*– and “Bovi”–vaccinated calves were significant (Figure 9B; Appendix A).

### 3.11. Both QMV–BVD2* and “Bovi” Vaccinated Calves Were Clinically Protected after BVDV–2 Challenge

After the challenge with BVDV2 until the day of euthanasia (20 dpc), clinical signs were recorded daily, based on the criteria listed in Appendix A. Rectal temperature (Appendix A), nasal and ocular discharge, lethargy, dyspnea, cough, mucosal or oral lesions, anorexia, and diarrhea were recorded to determine the clinical score for each calf (Appendix A). All animals from the unvaccinated control group developed a moderate (≥40 °C, 1/4 animals) to severe fever (≥40.5 °C, 4/5 animals) on or after 6 dpc, which lasted for several days (Figure 10A; Appendix A). Two calves in the “Bovi” group (# 627 and # 628) had mild (39.5–39.9 °C) to moderate (40.0–40.4 °C) non–specific fever during 3–6 dpc (Appendix A). Two calves (# 629 and # 632) had a mild non–specific fever during 0–5 dpc, and one calf (# 635) had a normal (38.5–39.2 °C) rectal temperature (Appendix A). In the QMV–BVD2* group, one calf (# 630), designated above as an outlier for nasal virus shedding and viremia (Section 3.9 and Section 3.10) had a moderate fever for three days (5–8 dpc), and another calf (# 640) had a moderate fever (40.0 °C ) on 6dpc but was normal the next day (Appendix A). The remaining two calves (# 633 and # 639) remained normal until the day of euthanasia.

Following the challenge, all control animals (5/5) also showed other clinical signs, in addition to fever, associated with the BVDV infection, i.e., nasal discharge, mild coughing, lethargy, anorexia, and diarrhea, which were scored. The mean clinical scores obtained for each of the three treatment groups following the challenge until 14 dpc are shown in Figure 10B. On 8 dpc, two control calves had mild to moderate nasal discharge, and the following day (9 dpc), all animals in the control group (5/5) had moderate to severe lethargy and anorexia. Two of the control animals also had diarrhea from 9 dpc until 13 dpc. In contrast, none of the calves vaccinated with QMV–BVD2* or “Bovi” commercial vaccine showed any of these signs noted above.

### 3.12. Based on Gross and Hispathological Lesions in the Lungs, the QMV–BVD2* Vaccine Protected the Calves Better than the “Bovi” Vaccine

#### 3.12.1. Gross Lesions

No gross lesions were found in the lungs of the QMV–BVD2* vaccines. Three of the five calves in control (sham–vaccinated) group had gross lesions. The control group’s lesions consisted of diffuse reddening and consolidation of the right cranial and cranial portion of the left cranial lung lobes in 2 calves (Figure 11). In the third calf, the cranioventral portions of the lungs were reddened, heavy and wet, but not consolidated. Two of the five “Bovi” vaccinees had detectable gross lesions. One (# 635) had bilateral reddening of the cranial lobes and the right middle lobe. There was a consolidation of individual lobules along the right middle and right cranial lung lobes along the ventral margins, with sparing of adjacent lobules (Figure 11). No consolidation was detected in the left lung. The second calf (# 628) had reddening of the right middle and bilateral cranial lobes in an irregular lobular pattern. Affected areas were heavy and wet, but minimally consolidated (Figure 11).

#### 3.12.2. Histopathology Findings

No significant differences were found among treatment groups in the histological scores of any tissues other than the lungs. The most consistent lesion in the lungs was an increase in peribronchial lymphocytes, either follicular or diffuse, with thick peribronchial cuffs in the most severely affected sections, especially in control and “Bovi” treatment groups. All groups had some degree of peribronchial fibrosis. All had inconsistent transmucosal neutrophilic exocytosis and some excess mucus within bronchi and/or bronchioles. However, only the “Bovi” and control groups had intraluminal neutrophils and rarely had bronchiolitis obliterans (2 of 5 in control and 1 of 5 in “Bovi”) (Figure 12 and Figure 13). These histopathological lesions, especially in bronchi/bronchioles, parenchyma and septa/pleura were scored individually for all lung sections in the three treatment groups (Appendix A). The most common parenchymal lesion seen in all groups was complete atelectasis of a portion or entire lobules, most likely caused by euthanasia. There were alveolar infiltrates in the “Bovi” and control groups, either histiocytic or mixed, neutrophilic, and histiocytic (Figure 12 and Figure 13). The infiltrates were limited and multifocal in the “Bovi” group (Figure 13) but more widespread in some sections from the control group (Figure 12). No alveolar infiltrates were present in the QMV–BVD2* group (Figure 14). Septal and/or pleural lesions of variable edema or fibrosis and mild lymphocytic infiltration were present in all groups.

Histological lesions seen in the other tissues were expected. Lymphoid hyperplasia with occasional early depletion was seen in most lymphoid organs. Sinus histiocytosis was often present. Minimal to mild lesions of chronic interstitial inflammation in the kidneys and minimal portal inflammation in the livers were common in all groups.

The statistical analysis of histological lesion scores recorded in Appendix A indicated that there were significant differences in the efficacy levels across the three groups (*p* < 0.049, $\eta_{p}^{2}$ = 0.42). Specifically, a significant difference was detected between the QMV–BVD2* and control groups in Bronchi (*p* < 0.05). In other words, QMV–BVD2*(M = 1, SD = 0.67) protected the calves significantly better for lung lesions after the BVDV–2 challenge when compared with the calves in the control group (M = 1.76, SD = 1.12) (Figure 15; Appendix A).

## 4. Discussion

Current vaccination practices against the viruses causing BRDC include trivalent attenuated, BoHV–1, BVDV–1 and –2 live vaccines. While these vaccines protect against the severity of BoHV–1 and BVDV infections, these vaccines were linked to outbreaks of abortion (BoHV–1) in dairy cattle industries, respiratory diseases (BoHV–1 and BVDV) in the beef and dairy cattle industries, and persistent infections (BVDV) in dairy cattle industries. In several cases, the causal agent(s) was traced back to the vaccine strain of BoHV–1 used in the polyvalent vaccine because the traditional BoHV–1–MLV vaccine virus establishes latency in the TG, reactivates with stress, and be shed in the nasal secretions. Therefore, only the gE–deleted BoHV–1 marker vaccine is allowed in several EU countries for vaccination against BoHV–1. The BoHV–1 gE–deleted marker vaccine is distinguishable from the BoHV–1 MLV strains serologically. Under field conditions, the gE marker vaccine virus in most cases was not shed from the nose of vaccinated animals following reactivation from the latency. However, a low–level gE marker virus shedding occurred in some instances of latency–reactivation (http://ec.europa.eu/food/fs/sc/scah/out49_en.pdf) [43].

The live attenuated BVDV strains used in the multivalent BRD vaccines are suspected in BVDV–associated problems in the cattle industry because of its RNA genome’s inherent ability to mutate under the field conditions. Additionally, like the wt BVDV, the vaccine virus also causes immunosuppression and vertical transmission in pregnant cows and persistent infection of calves [19,44,45,46,47]. Recently, the single Npro and double Npro–Erns live BVDV mutants were also developed to avoid the traditional BVDV MLV vaccine–associated problems. However, both the mutant viruses can cross the placental barrier and established persistent infection [19,48]. Therefore, traditional MLV and genetically engineered BVDV vaccines are not allowed in many EU countries or discouraged. Instead, eradication of BVDV by (i) testing and identifying newborn calves for persistent BVDV infection, (ii) removing PI calves, and (iii) taking hygienic, and biosecurity control measures have been implemented. However, this latter approach renders the naïve cattle population vulnerable to severe and widespread BVDV infection if the virus is introduced into the cattle population.

This study constructed a BoHV–1 quadruple mutant virus, BoHV–1QMV, which lacks the BoHV–1 UL49.5 Ecto–domain residues (30–32) plus the CT residues (80–96), the entire gE CT and Us9, and gG. Further, we inserted the chimeric BVDV–2 E2 and E^rns^–GM–CSF genes in the gE CT–Us9 and gG deletion loci, respectively. We determined that the resulting QMV–BVD2* vaccine elicited higher cross–reactive IFN–γ and proliferation responses in the vaccinated calves against BVDV–1 and –2 before and after the virulent BVDV–2 challenge when compared with the "Bovi"–vaccinated group. The QMV–BVD2* vaccine also induced BVDV–2–specific seroconversion at 34 dpv, but the “Bovi” generated a 50–fold higher BVDV–2 serum–neutralizing titer by that time. Nevertheless, QMV–BVD2* vaccinated calves had a better recall neutralizing antibody response after the challenge (6 dpc), and the BVDV–2–specific SN titer increased four–fold in the QMV–BVD2*–vaccinated calves (17 to 64). In contrast, the SN titers in the “Bovi” group remained the same (878 on 0 dpc and 885 on 6dpc). Notably, by 14dpc, the SN titer in the QMV–BVD2* sky–rocketed 5500–fold relative to that on the day of the challenge, whereas the corresponding increase in SN titer in the “Bovi” group was 19–fold. Taken together, upon BVDV–2 challenge, QMV–BVD2* vaccinated claves had stronger and higher levels of BVDV cross–reactive (types 1 and 2) T cell as well as BVDV–2–specific recall SN neutralizing responses compared with that of “Bovi” (commercial trivalent MLV). Most notably, the calves in both the control–unvaccinated and the “Bovi” vaccinated groups had visible gross lung–lesions. Consistent with this finding, the animals in the “Bovi” and control unvaccinated groups had also lung histopathological lesions; however, the QMV–BVD2* vaccinated calves had either mild or no lesion. Therefore, we believe that the QMV–BVD2* primed and induced a better BVDV–specific memory T cell response than that of “Bovi” (MLV). Consistent with this assumption, previous attempts to use BoHV–1 vectored BVDV subunit E2 vaccines were not adequately protective even though they induced BVDV–specific neutralizing antibody response [49,50]. Notably, these BoHV–1 vectors still had the two immunosuppressive genes, U_L_49.5 and gG, intact in their genome. Also, in those instances, only the BVDV E2 was used as a subunit antigen. In contrast, we additionally used BVDV2 Erns fused with the bovine GM–CSF chimeric protein as a second subunit antigen. 

Taken together, we believe that deleting both the immunosuppressive BoHV–1 genes in the vaccine vector combined with the inclusion of GM–CSF together with Erns most likely contributed towards improved cellular and memory neutralizing antibody responses against BVDV. Remarkably, even though the subunit antigens expressed by the QMV–BVD2* were type 2 BVDV–specific, the cellular immune response induced by the prototype vaccine was reactive against both BVDV–1 and –2. Earlier, we also determined that the BoHV–1 TMV was equally attenuated as a gE–deleted virus but induced a better protective immune response against the virulent BoHV–1 challenge compared with the gE–deleted virus with respect to both the cellular immune and neutralizing antibody responses. In the case of QMV–BVD2*, in which the BoHV–1 gG gene was additionally deleted, the efficiency of virus replication in the nasal mucosa was reduced slightly compared with that of TMV [28]. Nevertheless, BoHV–1 QMV induced slightly higher BoHV–1–specific neutralizing antibody response compared with that of BoHV–1 TMV [28]. While in this study, we did not determine its vaccine efficacy against BoHV–1, but based on its comparable neutralizing antibody response to that of "Bovi"–vaccinated animals, we believe that the QMV–BVD2* will be equally or better protective against BoHV–1 than the BoHV–1 TMV. Taken together, we believe the QMV–BVD2* vaccine is similarly or slightly better protective against BoHV–1, BVDV–1, and BVDV–2 compared with that of “Bovi” vaccine. By using the QMV–BVD2* vaccine, we will get comparable or equal protection against the three viruses and avoid the MLV BoHV–1 and BVDV vaccines associated problems in the field.

Further, from the manufacturing perspective, the vaccine would be cost–effective. This is because instead of growing the three different viruses (BoHV–1, BVDV–types 1 and 2) to formulate the vaccine, only the QMV–BVD2*, which grows with a much higher titer in the MDBK cells compared with that of BVDV, will provide the BVDV protective antigens. Furthermore, based on the gE CT–based marker assay [29], the QMV–BVD2* vaccinated animals can be distinguished from the wt BoHV–1–infected animals in the field. Also, a commercial BVDV NS3 based blocking ELISA test kit (BIO K 230; Bio-X Diagnostics, Rochefort, Nouvelle-Aquitaine, Belgium) exists. This could be used as a serological marker test to distinguish the QMV–BVD–2* vaccinated animals from the BVDV–infected animals. Therefore, QMV–BVD2* will fulfill the DIVA property against both BoHV–1 and BVDV to distinguish the vaccinated animals from the infected animals under field conditions.

## 5. Conclusions

Current polyvalent modified live vaccines against BoHV-1 and BVDV have limitations concerning their safety and efficacy. This study addressed the problem by demonstrating that a genetically engineered BoHV-1 QMV vectored subunit vaccine against BVDV 2 (QMV- BVD2*) is safe and efficacious against a virulent BVDV 2 challenge. Here, our results demonstrated that the QMV-BVD2* prototype subunit vaccine expressing chimeric BVDV-2 E2 and E^rns^ – fused with bovine GM-CSF induced BoHV-1- and BVDV-2-specific neutralizing antibody responses along with BVDV-1 and -2 cross-reactive cellular immune responses. Moreover, after a virulent BVDV-2 challenge, the QMV-BVD2* prototype subunit vaccine conferred a more rapid recall of BVDV-2-specific neutralizing antibody response and considerably better recall of BVDV-1 and -2-cross protective cellular immune responses than that of the Zoetis Bovi-shield Gold 3. Therefore, we suggest that the QMV-BVD2* subunit vaccine will be a safer but equally protective replacement for the current traditional trivalent vaccine against BoHV-1 and BVDV types 1 and 2.

## Figures and Tables

**Figure 1 vaccines-09-00046-f001:**
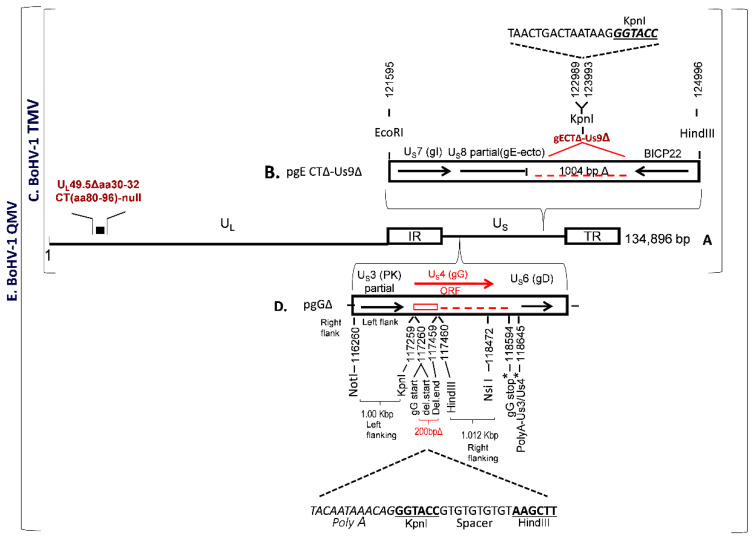
Schematic showing genomic configuration of BoHV–1 TMV constructed previously and the strategy of BoHV–1 gG gene deletion to generate a BoHV–1 QMV vaccine vector. (**A**) Schematic of BoHV–1 U_L_49.5Δ30–32 CT–null virus backbone [14]. U_L_ = unique long region; U_S_ = Unique short region; I_R_ = Internal repeat region and T_R_ = Terminal repeat region. (**B**) Plasmid construct pgE CTΔ–Us9Δ used to introduce CTΔ–Us9Δ in the BoHV–1 U_L_49.5Δ30–32 CT–null virus backbone (**A**) to generate a BoHV–1 triple mutant virus (BoHV–1 TMV) [28]. (**C**) Genomic configuration of BoHV–1 TMV. (**D**) Plasmid construct used to incorporate gG deletion in the BoHV–1 TMV to generate BoHV–1 QMV. Arrows indicate the direction of the corresponding open reading frame (ORF) and (**E**) Genomic organization of BoHV–1 QMV. Nucleotide numbers correspond to GenBank accession # JX898220.

**Figure 2 vaccines-09-00046-f002:**
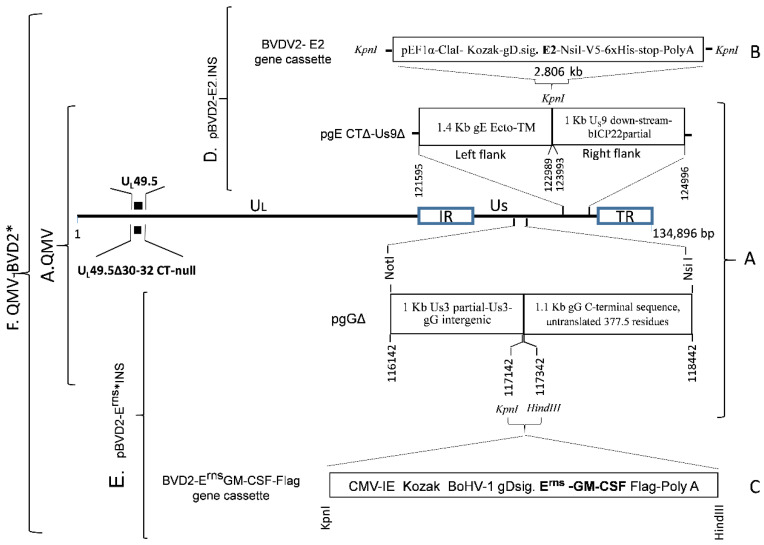
The strategy of chimeric BVD2 E2 and Erns–GM–CSF insertion in the gE CT–Us9 and gG deletion loci, respectively, of BoHV–1 QMV genome to generate BoHV–1 QMV–BVD2 E2–Erns–GM.CSF virus (QMV–BVD2*). (**A**) Schematic of BoHV–1 triple mutant virus (TMV) genomic organization showing unique long 49.5 (U_L_ 49.5), gE CT/Us9 and gG deletions. Nucleotide numbers correspond to GenBank accession # JX898220. (**B**) Schematic of BVDV2–E2 expression cassette. (**C**) Schematic of BVDV2–Erns–GMCSF. Flag expression cassette (Erns*). (**D**) BVD2–E2 expression cassette cloned into the KpnI site of pgE CTΔ–Us9Δ to yield pBVD2–E2.INS. (**E**) BVDV2–Erns* expression cassette cloned into the KpnI–HindIII restriction sites of pgGΔ to yield pBVD2–Erns* INS. (**F**) Incorporation of BVDV E2 and Erns* expression cassettes into the BoHV–1 QMV to yield BoHV–1 QMV–BVD2*.

**Figure 3 vaccines-09-00046-f003:**
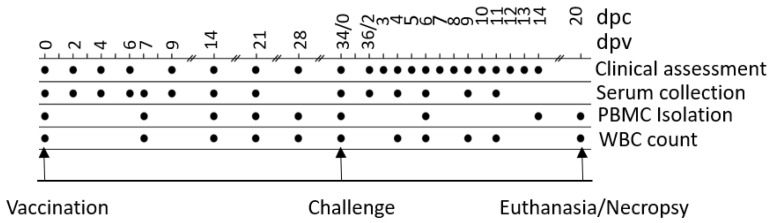
Vaccination, sample collection, challenge, and euthanasia scheme for the animal experiment. dpc: days post challenge; dpv: days post vaccination; PBMC: peripheral blood mononuclear cells; WBC: white blood cells.

**Figure 4 vaccines-09-00046-f004:**
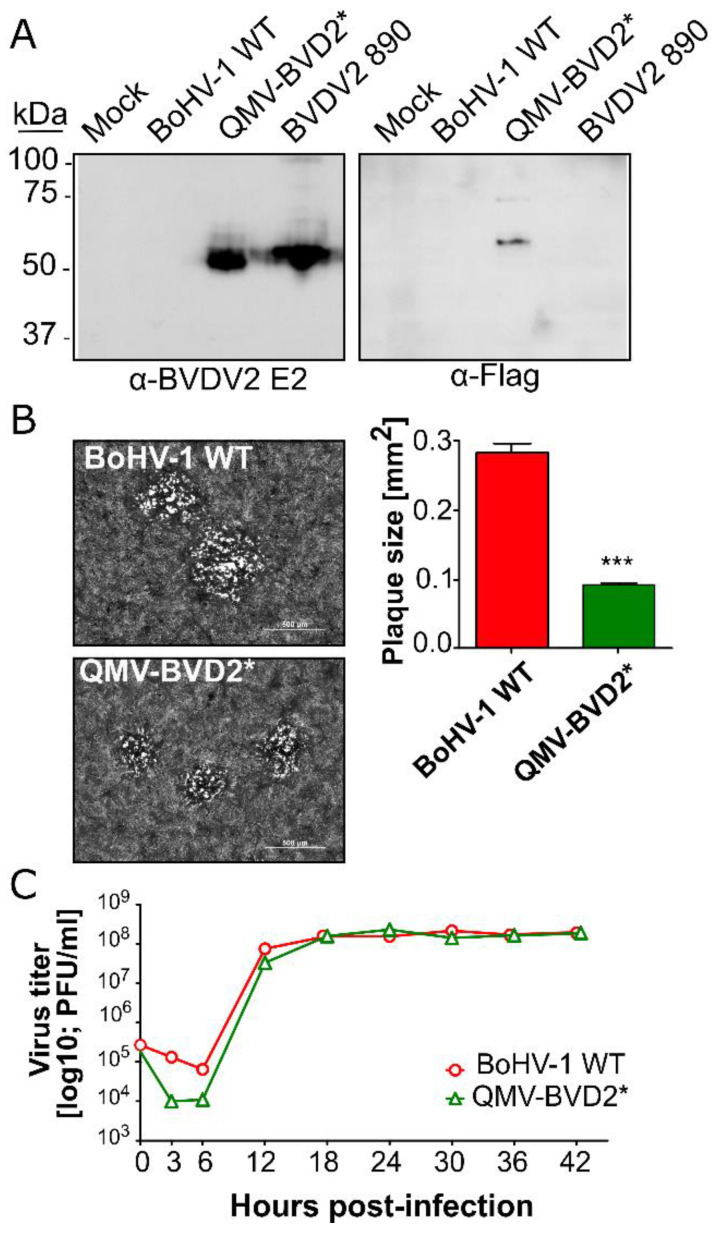
In vitro characterization of QMV–BVD2*. (**A**) Immunoblot analysis of QMV–BVD2* expressing chimeric BVDV2 E2 and Flag–tagged Erns–GM–CSF proteins by using anti BVDV2 E2–specific (left panel) and anti–Flag–specific (right panel) antibodies, respectively. (**B**) Plaque size analysis of QMV–BVD2* compared to that of BoHV–1 wt. Shown are the pictures of areas containing two–three representative plaques of each virus. The bar graph shows the average plaque size of at least 50 plaques with SD (*** *p* < 0.001). (**C**) One–step growth analysis of QMV–BVD2* compared with the BoHV–1 wt.

**Figure 5 vaccines-09-00046-f005:**
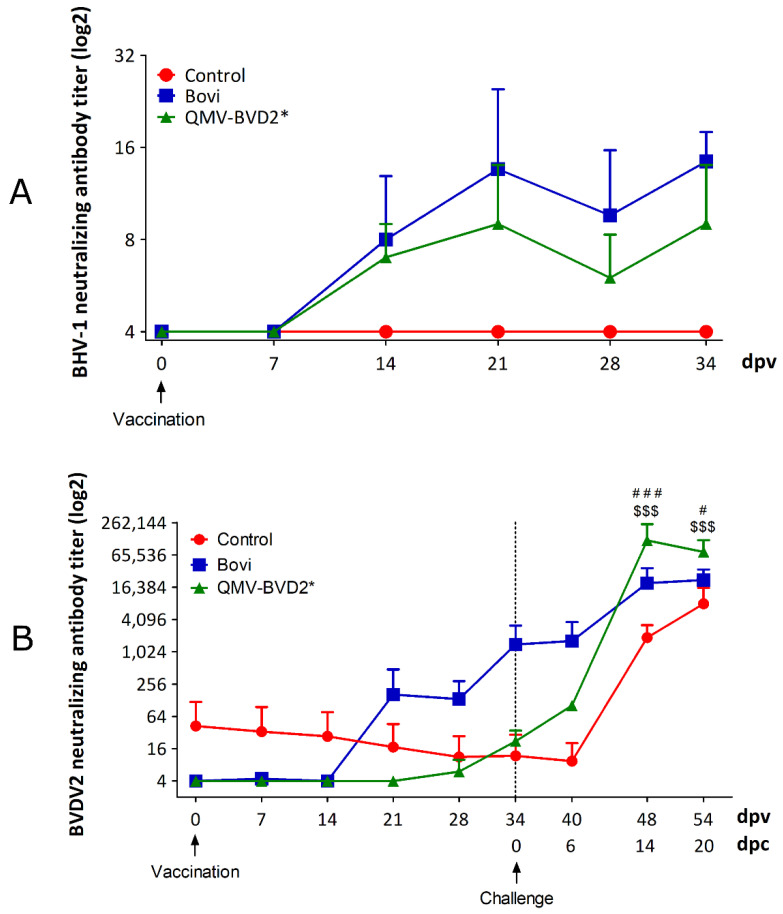
Serum virus neutralizing (SN) antibody titers after vaccination and challenge. SN antibody titers were determined by standard plaque reduction assay. (**A**) BoHV–1–specific SN antibody titer against BoHV–1 Cooper following vaccination. (**B**) BVDV–2–specific SN antibody titer against cp BVDV–2 125. Shown is the mean of each group with SD. Neutralizing titers obtained in each animal are shown in Appendix A. The data represent the mean + standard deviation. Significant differences in BVDV–2 neutralizing antibody titer were seen between different groups. Two–way ANOVA followed by Bonferroni post–tests to compare replicate means by row; $$$ *p* < 0.001 between Control and QMV–BVD2* group. # *p* < 0.05, ### *p* < 0.001 between Bovi and QMV–BVD2* group.

**Figure 6 vaccines-09-00046-f006:**
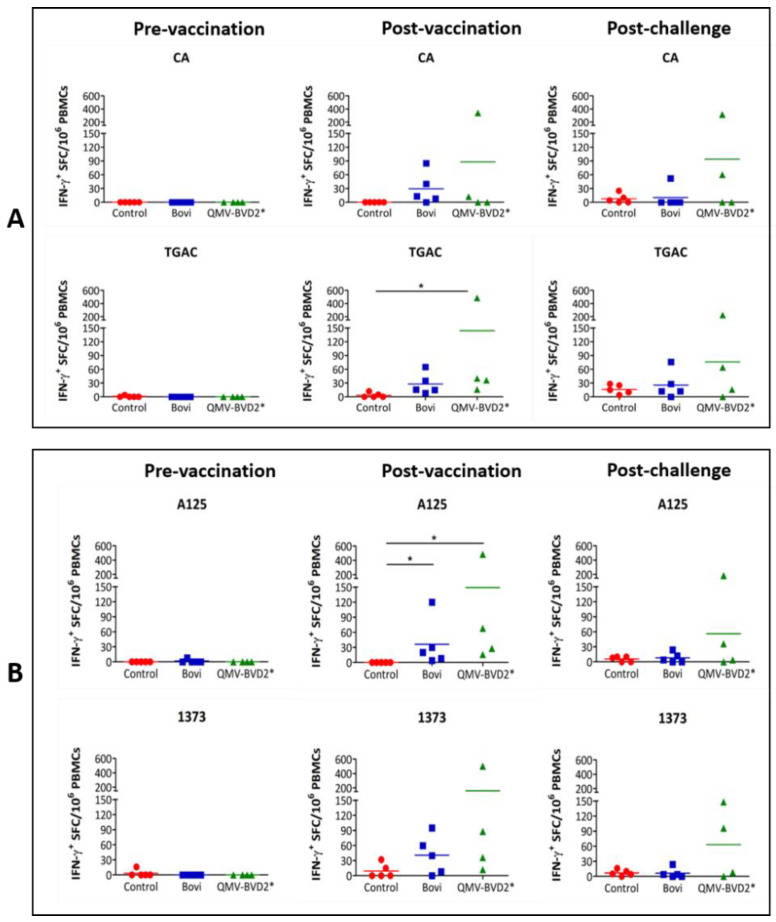
Pre–vaccination, Post–vaccination, and post–challenge BVDV–1 (**A**) and BVD2 (**B**) strain–specific IFN–γ cellular responses. IFN–γ secreting PBMC responses against BVDV–1 (CA and TAGC) and BVDV–2 (A125 and 1373) strains were determined by IFN–γ ELISPOT assays. The response is presented as IFN–γ^+^ SFC/10^6^ PBMCs. Medium alone served as the negative control, and the data shown are normalized by deducting media background counts. The group mean is represented by a bar. Asterisks denote statistically significant differences between the groups (* *p* < 0.05).

**Figure 7 vaccines-09-00046-f007:**
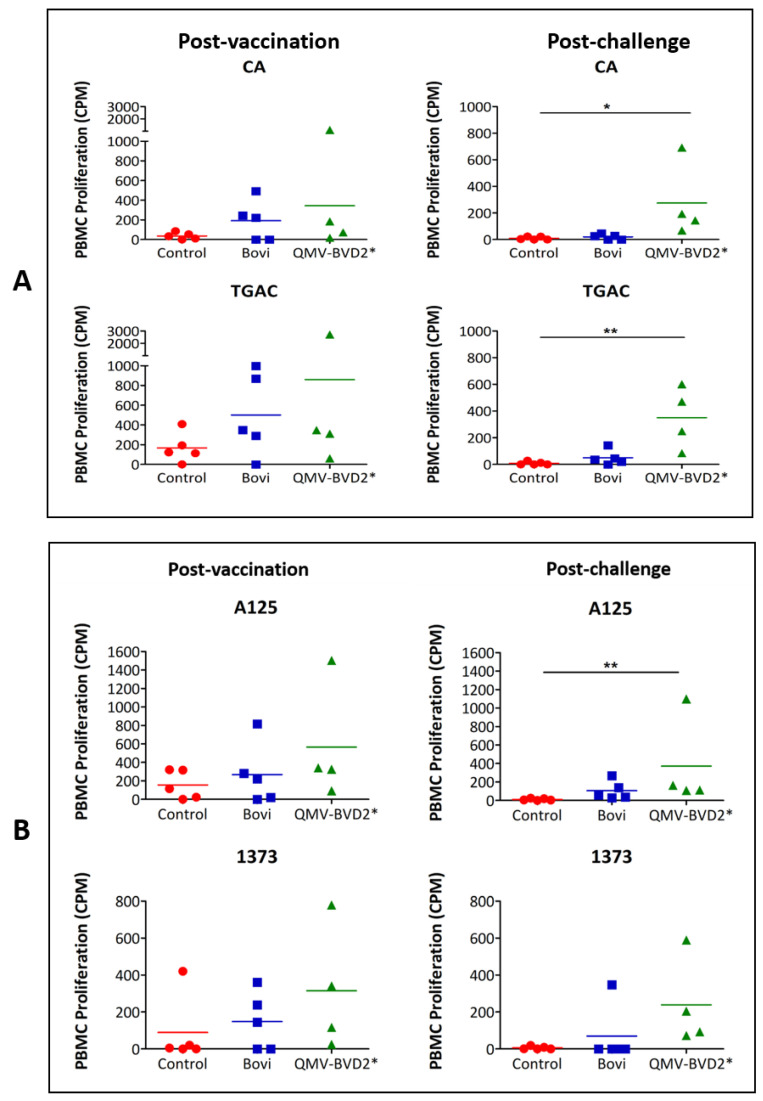
**Post–vaccination** and post–challenge BVDV–1 (**A**) and BVDV–2 (**B**) strain–specific proliferation of the PBMCs. PBMC responses against BVDV–1 and BVDV 2 strains were determined by cell proliferation assays. The incorporation of ^3^H–thymidine by the proliferating PBMCs is presented as CPM. Medium alone served as the negative control, and the data shown is minus media background counts. The group mean is represented by a bar. Asterisks denote statistically significant differences between the groups (* *p* < 0.05 and ** *p* < 0.01).

**Figure 8 vaccines-09-00046-f008:**
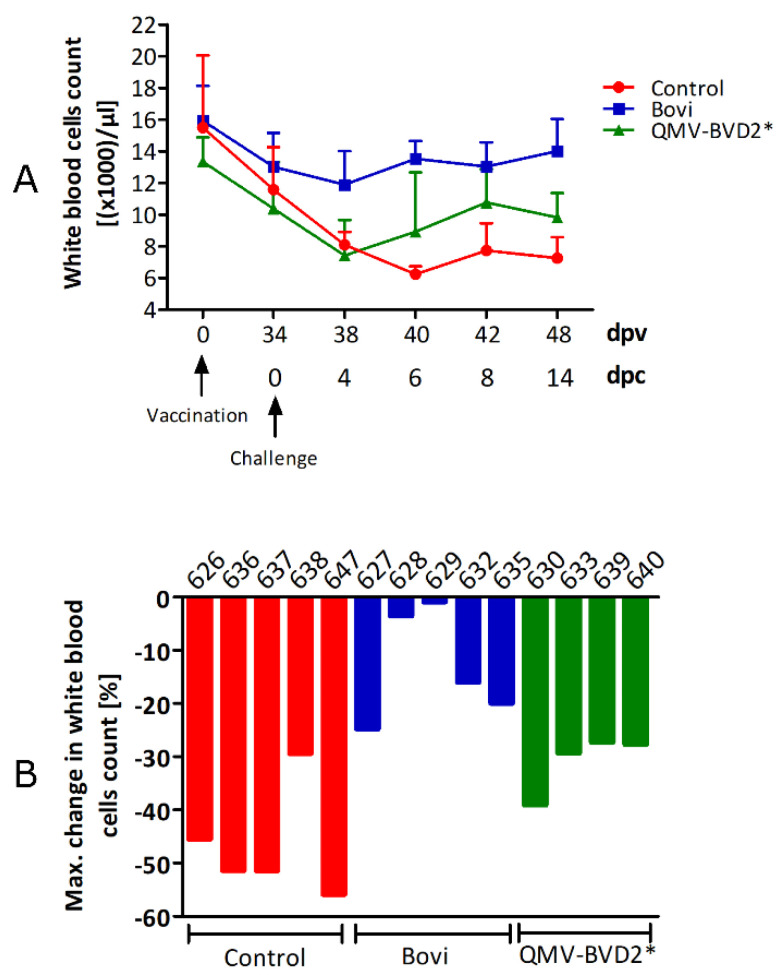
Leukocytopenia after challenge. (**A**) Shown is the mean of the leukocyte count for each treatment group with SD at the indicated days. (**B**) Shown is the mean comparative percent decline in leukocyte numbers for each animal of the 3 treatment groups.

**Figure 9 vaccines-09-00046-f009:**
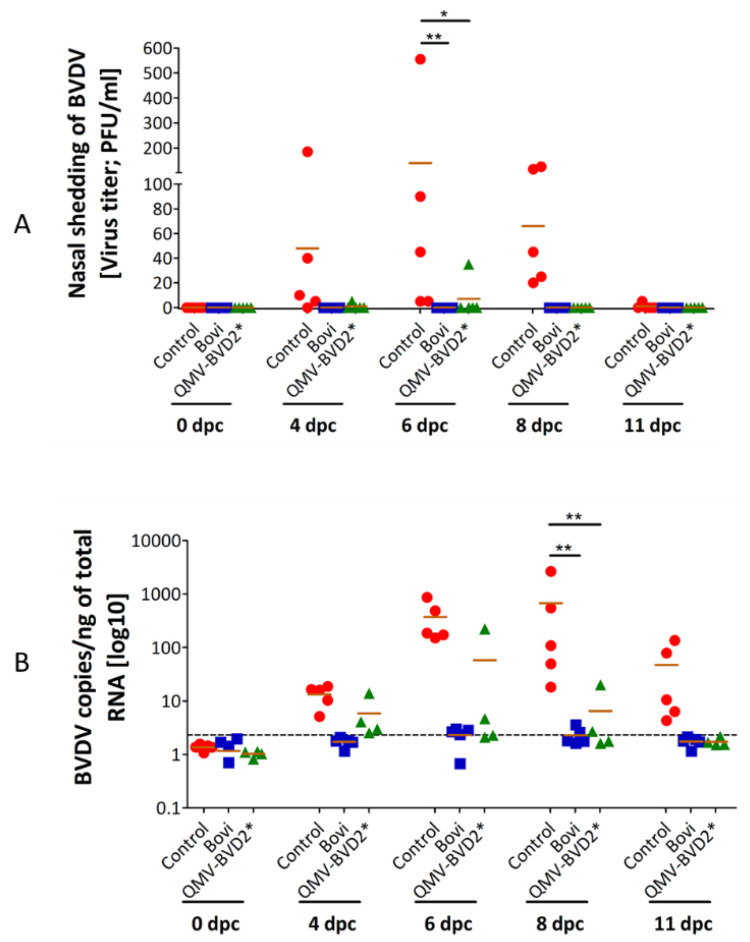
BVDV nasal virus shedding and viremia following challenge with BVDV–2 890 strain. (**A**) Virus isolated from each animal’s nasal swabs following the challenge with BVDV 2 non–cytopathic strain 890 was titrated in MDBK cells by plaque assay as described in the materials methods for the cytopathic strain 125. However, the viral plaques were visualized after 96 h post–inoculation using BVD2–E2–specific mAb and fluorescent–tagged secondary antibody. Shown are the titers of each animal of the three vaccination groups on 0 dpc, 4 dpc, 6 dpc, 8 dpc and 11 dpc. The data represent mean + individual values in each group. Significant differences in nasal shedding of BVDV virus titer were seen between different groups. n = 5 (QMV–BVD2* group contained 4 animals); Two–way ANOVA followed by Bonferroni post–tests to compare replicate means by row; * *p* < 0.05, ** *p* < 0.01. Grubb’s test (generalized extreme studentized deviate method) revealed the existence of outlier in BVD–QMV2* group (Animal # 630). On both 4 and 8 dpc, Calf #630 in BVD–QMV2* group was a significant outlier (*p* ˂ 0.05) for nasal virus shedding, in comparison to other calves in the same group (Appendix A). (**B**) BVDV viremia was assessed by Real–time quantitative RT–PCR using the VetMax–Gold BVDV kit (ThermoFisher, # 4413938) after the challenge with 890. BVDV2 genomic copy numbers were calculated according to CT–values of a standard curve. Shown is the mean copy number of the BVDV genome in one ng total RNA of three independent PCR analyses in duplicates of each animal of the three vaccination groups on 0 dpc, 4 dpc, 6dpc, 8 dpc and 11 dpc. The dotted line separates negative from positive results and is based on the highest copy number detected in samples from 0 dpc (Threshold of 2.32 copies/ng total RNA). The data represent mean + individual values in each group. Significant differences in BVDV genome copies were seen between different groups. n = 5 (QMV–BVD2* group contained 4 animals); Two–way ANOVA followed by Bonferroni post–tests to compare replicate means by row; ** *p* < 0.01. As above in “A”, according to Grubb’s test–generalized extreme studentized deviate method, on 4, 6 and 8 dpc, Calf # 630 in BVD–QMV2* group was a significant outlier for viremia (*p* ˂ 0.05) in comparison to other calves in the group (Appendix A).

**Figure 10 vaccines-09-00046-f010:**
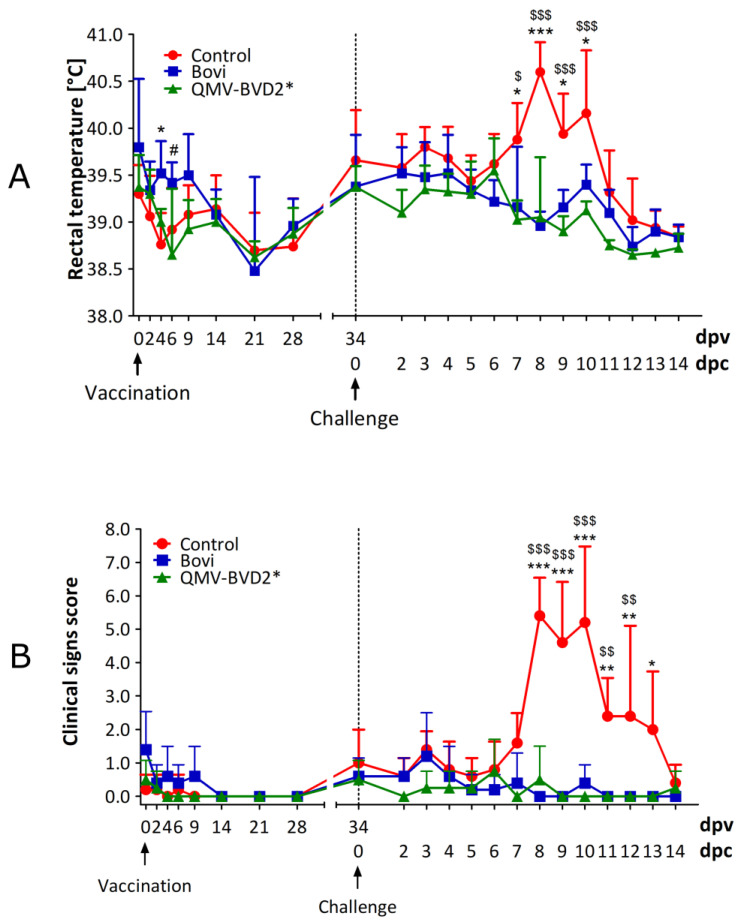
Clinical assessments. (**A**). Rectal temperature of calves following vaccination and challenge. Rectal temperature was measured with a digital thermometer on the indicated days. Shown is the mean temperature of each treatment group with SD. The data represent the mean + standard deviation. Significant differences in rectal temperature were seen between different groups. n = 5 (QMV–BVD2* group contained 4 animals); Two–way ANOVA followed by Bonferroni post–tests to compare replicate means by row; * *p* < 0.05, *** *p* < 0.001 between Control and Bovi group. # *p* < 0.05 between Bovi and QMV–BVD2* group. $ *p* < 0.05, $$$ *p* < 0.001 between Control and QMV–BVD2* group. (**B**). Clinical scores in calves of each treatment group following vaccination and after BVDV–2 challenge is shown. According to a scoring system reported earlier [42], minor modifications are listed in Appendix A. Shown is the mean of each group with SD. Significant differences in clinical signs score were seen between different groups. n = 5 (QMV–BVD2* group contained 4 animals); Two–way ANOVA followed by Bonferroni post–tests to compare replicate means by row; * *p* < 0.05, ** *p* < 0.01, *** *p* < 0.001 between Control and Bovi group. $$ *p* < 0.01, $$$ *p* < 0.001 between Control and QMV–BVD2* group.

**Figure 11 vaccines-09-00046-f011:**
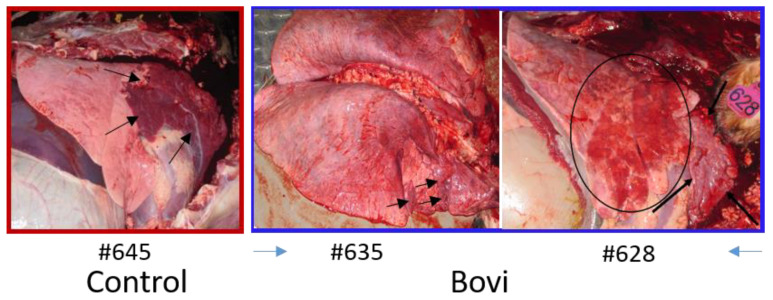
Gross pathology of lung. Lungs from calf # 645 (control group). Cranioventral consolidation of the lungs in the area outlined by arrows. This portion is consolidated with a classical pattern of acute bronchopneumonia. Lungs from calf # 635 (“Bovi” group): Note small areas of reddening and consolidation (arrows) in the cranioventral area of the right middle and cranial lobes. Lungs from calf # 628 (“Bovi” group): Arrows delineate extensively consolidated and fibrotic right cranial lung lobe. The circled area is irregularly reddened. Histologically the latter area had mild, multifocal acute inflammation.

**Figure 12 vaccines-09-00046-f012:**
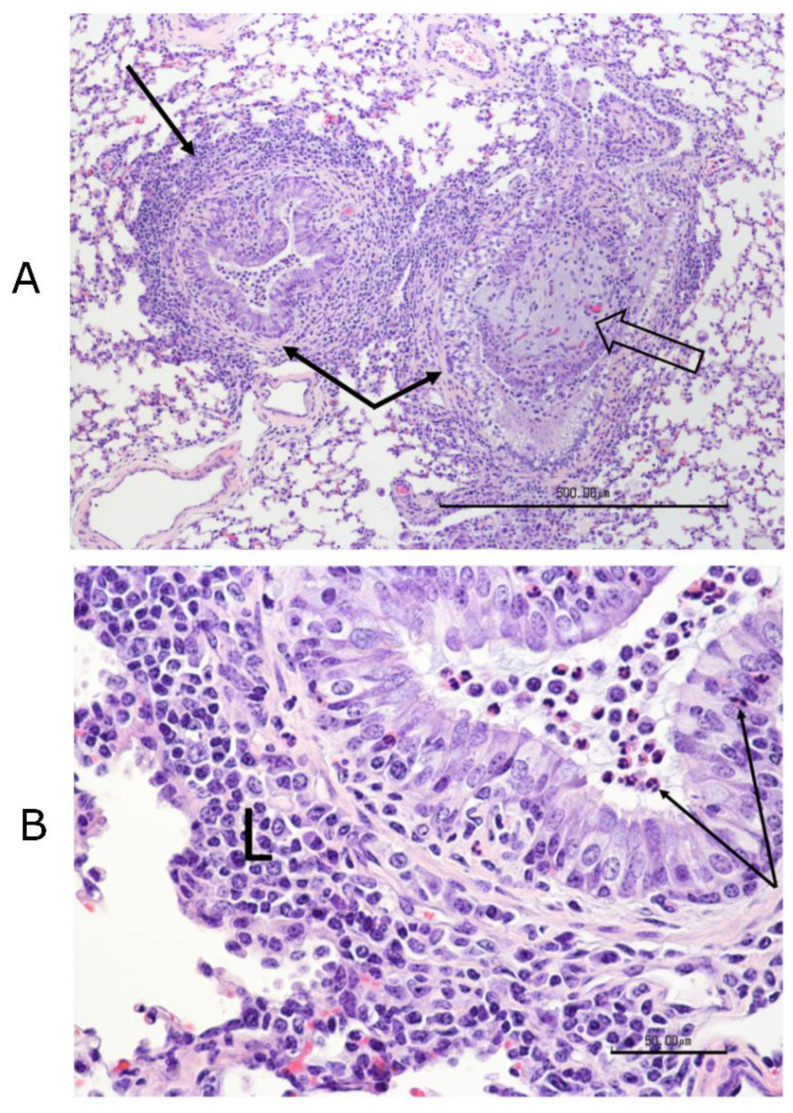
Histopathology of lung sections. Representative sections of lung tissues showing lesions from control (Figure 12), “Bovi” (Figure 13) and QMV–BVD2* (Figure 14) vaccinated groups. Figure 12 (**A**) Control group calf: The lung has substantial bronchiolar damage. The open arrow shows fibrosis filling the lumen of the affected bronchiole (bronchiolitis obliterans). A lymphoplasmacytic cuff surrounding the bronchioles (single solid arrow) and smooth muscle hypertrophy and fibrosis (double arrows). Bar = 500 µM (**B**) Control group calf: The bronchiole is surrounded by a cuff of mononuclear cells (L), plasma cells, lymphocytes, and fewer macrophages. Neutrophilic transmucosal exocytosis with accumulation in the bronchiolar lumen (double arrows) indicates that the inflammation remains active bar = 50 µM.

**Figure 13 vaccines-09-00046-f013:**
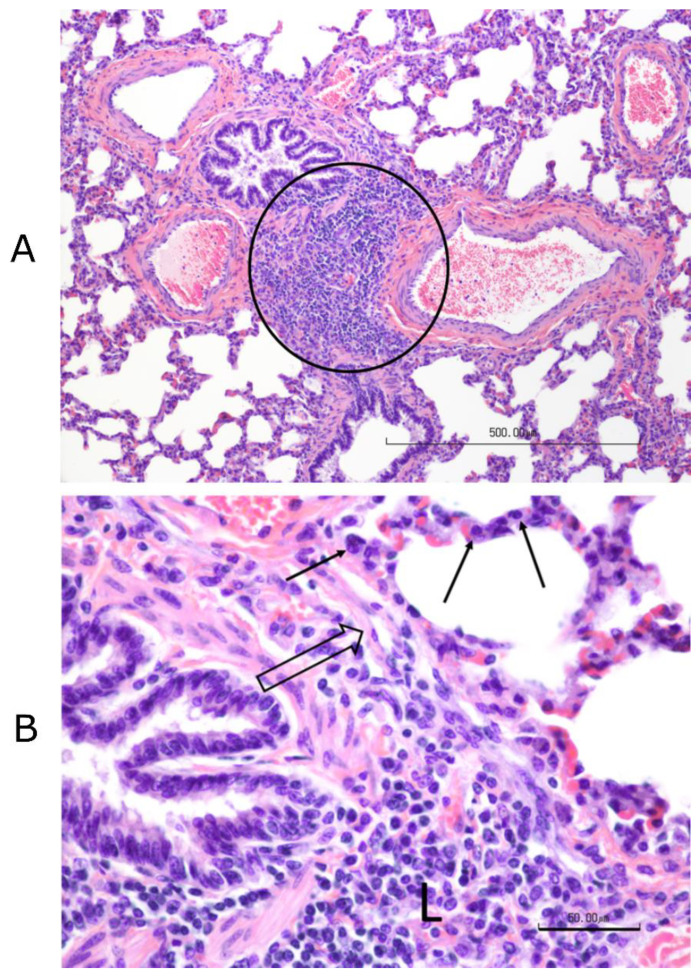
(**A**) Commercial “Bovi” vaccine group: Notice the marked increase in BALT: bronchus–associated lymphoid tissues (encircled). The bronchiole is constricted, which is not considered a significant lesion. Alveolar spaces are clear. Bar = 500 µM. (**B**) “Bovi” vaccine group: Marked increase in BALT (L). Mild fibrosis is evident (open arrow). A few mononuclear inflammatory cells have infiltrated the adjacent alveolar septa (solid arrows). Bar = 50 µM.

**Figure 14 vaccines-09-00046-f014:**
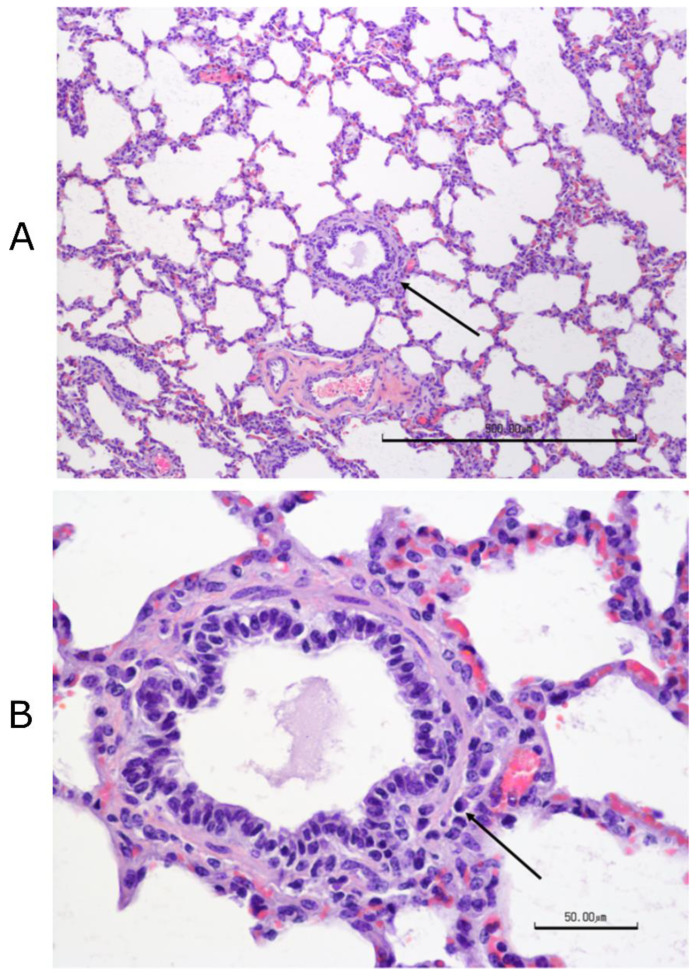
(**A**) QMV–BVD2* vaccine group: Minimal bronchiolar change with few peribronchiolar lymphocytes and plasma cells (arrow). Alveolar spaces are clear, and septa are normal. Bar = 500 µM. (**B**) QMV–BVD2* vaccine group: Limited plasma cells (arrow) and lymphocytes are present in the peribronchiolar interstitium in approximately normal numbers. The smooth muscle may be mildly hypertrophied, but it also may be an artifact of perimortem contraction. Bar = 50 µmM.

**Figure 15 vaccines-09-00046-f015:**
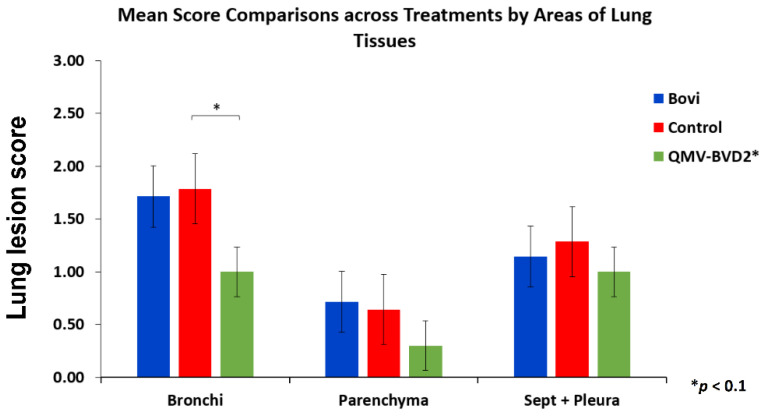
Mean histopathology–lesion scores across treatment groups by areas of lung tissues. Note that a significant difference was detected between the QMV–BVD2* and control groups in Bronchi (* *p* < 0.1). In other words, QMV–BVD2*(M = 1, SD = 0.67) protected the calves significantly better from the lung lesions after the BVDV–2 challenge when compared to the calves in the control group (M = 1.76, SD = 1.12).

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
