# Peer review of "BoHV-1-Vectored BVDV-2 Subunit Vaccine Induces BVDV Cross-Reactive Cellular Immune Responses and Protects against BVDV-2 Challenge"

_vaccines, 2021, doi:10.3390/vaccines9010046_

Round 1
Reviewer 1 Report
Co-authors Shafiqul I. Chowdhury and Katrin Pannhorst and collaborators present here a research manuscript where they evaluate the protective efficacy of a vaccine against BHV1 and BVDV2 in calves after challenge with both viruses. Authors seem to be aware of the status of the vaccination that is widely used to produce herd immunity with both inactivated and live attenuated against BHV1 and BVDV2, as they mentioned and later compared to the commercially available polyvalent modified live vaccine. However, as they point out, these vaccines have limitations concerning safety and protective efficacy. So, they generated a quadruple gene-edited BHV1 that also expresses BVDV and provides information about the immune response in calves. In my opinion, the manuscript is well-written, the experimental design is clean and well-planned, the methods and genetic engineer constructs are beautifully detailed. The authors suggest here that their engineered vaccine is substantially better because provides a strong cellular immune response based on the neutralizing antibodies analysis and reduced virus shedding. Altogether the results presented here will benefit not only the field of immunology but also will potentially have an impact on the cattle industry. However, it requires extensive corrections detailed below:
Major:
- Figure 1A. Appropriate loading control is critical for this Western blot analysis. Usually, a stripped membrane can be re-blotted against housekeeping proteins, such as β-actin, glyceraldehyde-3-phosphate dehydrogenase, or and β-tubulin, to normalize protein expression. Also, the expression of the selected protein needs to show no impact under their experimental conditions.
Why is the QMV-BVD2 a-Flag different than the a-BVDV2-E2, if is glycosylated, does it supposed to be glycosylated in both blots? Why a-Flag shows a double band of higher molecular weight?
- Figure 1C. Explain why and how virus titer decreased after 3 and 6 hours post-infection.
- Figure 5B. Explain why the control group develops neutralizing antibodies if no immunization was performed and the colostrum effect was removed.
Minor:
- Please avoid acronyms in the titer. Also, I am not sure if the acronym according to the ICTV should be BHV1 or BoHV1, please correct accordingly. I think under the new nomenclature is BoHV1 [AJ004801=NC_001847] if this is the case, please correct it in the entire manuscript and be consistent, in this version line 280 says “BoHV-1 wt” under BHV-1 wt appendix.
- If line 36 to 38 is using the same reference [1], just put it at the end of line 38.
- Typo in reference 1, please remove the extra period.
- Line 40. Typo please remove Approx. from “approximaletly (Approx.)”
- Line 122-125. Is there any reason why they did not add L-glutamate to the DMEM?
- Why some samples even 0 dpc have a Ct value?
- Figure 5 and 8A. Please add the standard deviation in both + and –
- Figure 15. Please add labels on y-axes
Author Response
Comments and Suggestions for Authors (Reviewer # 1)
Co-authors Shafiqul I. Chowdhury and Katrin Pannhorst and collaborators present here a research manuscript where they evaluate the protective efficacy of a vaccine against BHV1 and BVDV2 in calves after challenge with both viruses. Authors seem to be aware of the status of the vaccination that is widely used to produce herd immunity with both inactivated and live attenuated against BHV1 and BVDV2, as they mentioned and later compared to the commercially available polyvalent modified live vaccine. However, as they point out, these vaccines have limitations concerning safety and protective efficacy. So, they generated a quadruple gene-edited BHV1 that also expresses BVDV and provides information about the immune response in calves. In my opinion, the manuscript is well-written, the experimental design is clean and well-planned, the methods and genetic engineer constructs are beautifully detailed. The authors suggest here that their engineered vaccine is substantially better because provides a strong cellular immune response based on the neutralizing antibodies analysis and reduced virus shedding. Altogether the results presented here will benefit not only the field of immunology but also will potentially have an impact on the cattle industry. However, it requires extensive corrections detailed below:
Major:
- Figure 1A. Appropriate loading control is critical for this Western blot analysis. Usually, a stripped membrane can be re-blotted against housekeeping proteins, such as β-actin, glyceraldehyde-3-phosphate dehydrogenase, or and β-tubulin, to normalize protein expression. Also, the expression of the selected protein needs to show no impact under their experimental conditions.
Response: In the revised manuscript, lines 273-287, we clarified how the infected cell lysates were prepared to normalize the amount of protein loaded, on the SDS-PAGE, between the different samples. As noted in the revised results section, lines 457-477, BVDV Erns protein is secreted as well as membrane-anchored. Therefore, using house-keeping proteins as a loading control may not be suitable in this case.
We determined that after several passages of the QMV-BVD2* virus, the expression of chimeric E2 and Flag-tagged Erns-GM-CSF proteins were stable under experimental conditions (lines 269-271).
Why is the QMV-BVD2 a-Flag different than the a-BVDV2-E2, if is glycosylated, does it supposed to be glycosylated in both blots? Why a-Flag shows a double band of higher molecular weight?
Response: Antibody against Erns is not commercially available. Therefore, we used the commercially available antibody against Flag to detect the chimeric Flag-tagged Erns-GM-CSF protein. The molecular mass of the two chimeric proteins immunoblotted with E2 and Flag-specific antibodies is different.
- Figure 1C. Explain why and how virus titer decreased after 3 and 6 hours post-infection.
Response: Usually, due to the eclipse phase, virus titers during 3-6 hours may decrease. However, this may vary depending on the mutant. Therefore, this is not unusual.
- Figure 5B. Explain why the control group develops neutralizing antibodies if no immunization was performed and the colostrum effect was removed.
Response: It was not easy to find calves without maternal antibody. However, in our study, we obtained calves from pregnant cows vaccinated with an inactivated BVDV vaccine. To avoid the interference of maternal antibodes with vaccination, calves with more than 4 neutralizing maternal antibody titers were placed in the control group. The rationale was that the titers will gradually go down further prior to the challenge. However, only one control calf (# 647) had a higher titer on the day of vaccination and even on the day of challenge (32 ), which contributed towards the mean titer. There was no increase in serum neutralizing titer in any of the control calves at any time before day 6 after the challenge (Supplementary Table 2). Therefore, the statement made above by the reviewer is not correct.
Minor:
- Please avoid acronyms in the titer. Also, I am not sure if the acronym according to the ICTV should be BHV1 or BoHV1, please correct accordingly. I think under the new nomenclature is BoHV1 [AJ004801=NC_001847] if this is the case, please correct it in the entire manuscript and be consistent, in this version line 280 says “BoHV-1 wt” under BHV-1 wt appendix.
Response: We changed the acronym to BoHV-1.
- If line 36 to 38 is using the same reference [1], just put it at the end of line 38.
- Response: Deleted.
- Typo in reference 1, please remove the extra period.
- Line 40. Typo please remove Approx. from “approximaletly (Approx.)” Corrected
- Line 122-125. Is there any reason why they did not add L-glutamate to the DMEM?
Response: There is no reason.
- Why some samples even 0 dpc have a Ct value?
- Response: We have used VetMax-Gold BVDV detection kit (ThermoFisher, MA, USA, #4413938) for detection of BVDV viremia in calves. cDNA was generated from 250 ng of total RNA and RT-qPCR was performed with 40 amplification cycles. In RT-qPCR, the cutoff Ct value was set to ˂38 as recommended by the manufacturer, because non-specific ampilification signals can be detected in subsequent cycles. Further, the RT-qPCR detection limit was32 copies/ng total RNA (Line: 381-384). In our experiment, Ct values on 0 dpc in all animals were higher than 38, which was due to non-specific amplification. Further, BVDV genome copy numbers calculated from these animals were much lower than the threshold detection limit (˂2.32 copies/ng total RNA). Therefore we consider the animal were negative for BVDV viremia of 0 dpc as indicated by the threshold line in Figure 9B.
- Figure 5 and 8A. Please add the standard deviation in both + and –
- Response: We chose not to include negative (-) standard deviation to avoid the overlap between the data points. However, data from the individual animals are listed in supplementary tables 2 and 3, respectively.
- Figure 15. Please add labels on y-axes
Comments and Suggestions for Authors (Reviewer # 1)
Co-authors Shafiqul I. Chowdhury and Katrin Pannhorst and collaborators present here a research manuscript where they evaluate the protective efficacy of a vaccine against BHV1 and BVDV2 in calves after challenge with both viruses. Authors seem to be aware of the status of the vaccination that is widely used to produce herd immunity with both inactivated and live attenuated against BHV1 and BVDV2, as they mentioned and later compared to the commercially available polyvalent modified live vaccine. However, as they point out, these vaccines have limitations concerning safety and protective efficacy. So, they generated a quadruple gene-edited BHV1 that also expresses BVDV and provides information about the immune response in calves. In my opinion, the manuscript is well-written, the experimental design is clean and well-planned, the methods and genetic engineer constructs are beautifully detailed. The authors suggest here that their engineered vaccine is substantially better because provides a strong cellular immune response based on the neutralizing antibodies analysis and reduced virus shedding. Altogether the results presented here will benefit not only the field of immunology but also will potentially have an impact on the cattle industry. However, it requires extensive corrections detailed below:
Major:
- Figure 1A. Appropriate loading control is critical for this Western blot analysis. Usually, a stripped membrane can be re-blotted against housekeeping proteins, such as β-actin, glyceraldehyde-3-phosphate dehydrogenase, or and β-tubulin, to normalize protein expression. Also, the expression of the selected protein needs to show no impact under their experimental conditions.
Response: In the revised manuscript, lines 273-287, we clarified how the infected cell lysates were prepared to normalize the amount of protein loaded, on the SDS-PAGE, between the different samples. As noted in the revised results section, lines 457-477, BVDV Erns protein is secreted as well as membrane-anchored. Therefore, using house-keeping proteins as a loading control may not be suitable in this case.
We determined that after several passages of the QMV-BVD2* virus, the expression of chimeric E2 and Flag-tagged Erns-GM-CSF proteins were stable under experimental conditions (lines 269-271).
Why is the QMV-BVD2 a-Flag different than the a-BVDV2-E2, if is glycosylated, does it supposed to be glycosylated in both blots? Why a-Flag shows a double band of higher molecular weight?
Response: Antibody against Erns is not commercially available. Therefore, we used the commercially available antibody against Flag to detect the chimeric Flag-tagged Erns-GM-CSF protein. The molecular mass of the two chimeric proteins immunoblotted with E2 and Flag-specific antibodies is different.
- Figure 1C. Explain why and how virus titer decreased after 3 and 6 hours post-infection.
Response: Usually, due to the eclipse phase, virus titers during 3-6 hours may decrease. However, this may vary depending on the mutant. Therefore, this is not unusual.
- Figure 5B. Explain why the control group develops neutralizing antibodies if no immunization was performed and the colostrum effect was removed.
Response: It was not easy to find calves without maternal antibody. However, in our study, we obtained calves from pregnant cows vaccinated with an inactivated BVDV vaccine. To avoid the interference of maternal antibodes with vaccination, calves with more than 4 neutralizing maternal antibody titers were placed in the control group. The rationale was that the titers will gradually go down further prior to the challenge. However, only one control calf (# 647) had a higher titer on the day of vaccination and even on the day of challenge (32 ), which contributed towards the mean titer. There was no increase in serum neutralizing titer in any of the control calves at any time before day 6 after the challenge (Supplementary Table 2). Therefore, the statement made above by the reviewer is not correct.
Minor:
- Please avoid acronyms in the titer. Also, I am not sure if the acronym according to the ICTV should be BHV1 or BoHV1, please correct accordingly. I think under the new nomenclature is BoHV1 [AJ004801=NC_001847] if this is the case, please correct it in the entire manuscript and be consistent, in this version line 280 says “BoHV-1 wt” under BHV-1 wt appendix.
Response: We changed the acronym to BoHV-1.
- If line 36 to 38 is using the same reference [1], just put it at the end of line 38.
- Response: Deleted.
- Typo in reference 1, please remove the extra period.
- Line 40. Typo please remove Approx. from “approximaletly (Approx.)” Corrected
- Line 122-125. Is there any reason why they did not add L-glutamate to the DMEM?
Response: There is no reason.
- Why some samples even 0 dpc have a Ct value?
- Response: We have used VetMax-Gold BVDV detection kit (ThermoFisher, MA, USA, #4413938) for detection of BVDV viremia in calves. cDNA was generated from 250 ng of total RNA and RT-qPCR was performed with 40 amplification cycles. In RT-qPCR, the cutoff Ct value was set to ˂38 as recommended by the manufacturer, because non-specific ampilification signals can be detected in subsequent cycles. Further, the RT-qPCR detection limit was32 copies/ng total RNA (Line: 381-384). In our experiment, Ct values on 0 dpc in all animals were higher than 38, which was due to non-specific amplification. Further, BVDV genome copy numbers calculated from these animals were much lower than the threshold detection limit (˂2.32 copies/ng total RNA). Therefore we consider the animal were negative for BVDV viremia of 0 dpc as indicated by the threshold line in Figure 9B.
- Figure 5 and 8A. Please add the standard deviation in both + and –
- Response: We chose not to include negative (-) standard deviation to avoid the overlap between the data points. However, data from the individual animals are listed in supplementary tables 2 and 3, respectively.
- Figure 15. Please add labels on y-axes- added.
Reviewer 2 Report
Manuscript written by Shafigul I., and et all is very well organized,particular for clinical assessments, considering that vaccination was done on calves. Development of novel QMV-BVD2 vaccine virus is very well described with all information about cloning procedures.I suggest to provide more information about characterization of this novel virus: more western blots with anti-viral proteins antibodies. Authors showed that vaccination with novel QMV-BVD2 virus induced high titer of viral neutralising antibodies,provided by proliferation of strain specific PBMCs and IFN-Ȳ secreting PBMC response. One of the major markers of immune protection for vaccine based on alphaherpes viruses are antibody-dependent cellular cytotoxicity (ADCC). If authors have reagents for this type of assay I suggest to perform some ADCC assay using serum from vaccinated calves.
Some specific comments:
1. Figure 4 demonstrate characterization of QMV-BVD by showing the western blot with anti-α-BVDV2-E2 and anti –Flag Abs. only. For better characterization of very interesting novel virus I suggest to perform western blot with Abs against of viral BHV proteins.
2. Antibodies to viruses produced after infection or vaccination can protect the host by virus neutralization or through antibody-dependent cellular cytotoxicity (ADCC). Authors showed convincible that vaccination with novel QMV-BVD2* induced high titer of BHV 1 and BVDV viral neutralizing Abs. Activation of cellular immune response was shown by IFN-ᵞ secreting PBMC response against BVDV-1 and BVDV-2 by ELISPOT assay and by strain –specific proliferation of the PBMCs. If authors have necessary reagents, I suggest to perform some ADCC assay serum of calves vaccinated by QMV-BVDV2 and "BOVI" vaccines.
Author Response
Response to Reviewer # 2 comments
Manuscript written by Shafigul I., and et all is very well organized,particular for clinical assessments, considering that vaccination was done on calves. Development of novel QMV-BVD2 vaccine virus is very well described with all information about cloning procedures.I suggest to provide more information about characterization of this novel virus: more western blots with anti-viral proteins antibodies. Authors showed that vaccination with novel QMV-BVD2 virus induced high titer of viral neutralising antibodies,provided by proliferation of strain specific PBMCs and IFN-Ȳ secreting PBMC response. One of the major markers of immune protection for vaccine based on alphaherpes viruses are antibody-dependent cellular cytotoxicity (ADCC). If authors have reagents for this type of assay I suggest to perform some ADCC assay using serum from vaccinated calves.
Response: Currently, we don’t have the reagents necessary for the ADCC assay. However, we showed the protective immune response by demonstrating neutralizing antibody and cellular immune responses. More importantly, our data showed that the memory neutralizing antibody response after the challenge was significantly better than the commercial vaccine.
Some specific comments:
- Figure 4 demonstrate characterization of QMV-BVD by showing the western blot with anti-α-BVDV2-E2 and anti –Flag Abs. only. For better characterization of very interesting novel virus I suggest to perform western blot with Abs against of viral BHV proteins.
Response. We have previously characterized the BoHV-1 TMV (Vaccine 32 (2014): 4905-4915), the parental virus of BoHV-1 QMV. The BoHV-1 QMV contains an additional deletion of the gG gene sequence. We determined the sequences flanking the gG gene along with the chimeric BVDV2 gene sequences in the QMV-BVD2* recombinant virus. Our primary goal in this manuscript was to determine that the BoHV-1 vectored subunit BVDV-2 vaccine protects against the virulent BVDV-2 challenge. We have documented that the QMV vector virus replicated efficiently in vitro in MDBK cells and also replicated in the nasal epithelium of vaccinated animals and induced a protective immune response against BVDV-2.
- Antibodies to viruses produced after infection or vaccination can protect the host by virus neutralization or through antibody-dependent cellular cytotoxicity (ADCC). Authors showed convincible that vaccination with novel QMV-BVD2* induced high titer of BHV 1 and BVDV viral neutralizing Abs. Activation of cellular immune response was shown by IFN-ᵞ secreting PBMC response against BVDV-1 and BVDV-2 by ELISPOT assay and by strain –specific proliferation of the PBMCs. If authors have necessary reagents, I suggest to perform some ADCC assay serum of calves vaccinated by QMV-BVDV2 and "BOVI" vaccines.
Response: Please see above.
Round 2
Reviewer 1 Report
Authors address all major concerns in this new version of the manuscript. In my opinion the document is ready for publication in vaccines.